# Dynamical Properties of Tokens in Self-Attention and Effects of Positional Encoding

**Duy-Tung Pham**[*]
FPT Software AI Center
Hanoi, Vietnam
tungpd10@fpt.com

**An Nguyen The**[*]
FPT Software AI Center
Hanoi, Vietnam
annt68@fpt.com

**Viet-Hoang Tran**
Department of Mathematics
National University of Singapore
hoang.tranviet@u.nus.edu

**Nhan-Phu Chung**
University of Economics Ho Chi Minh City
Ho Chi Minh City, Vietnam
phucn@ueh.edu.vn

**Xin T. Tong**
Department of Mathematics
National University of Singapore
mattxin@nus.edu.sg

**Tan M. Nguyen**[†]
Department of Mathematics
National University of Singapore
tanmn@nus.edu.sg

**Thieu N. Vo**[†]
Department of Mathematics
National University of Singapore
thieuvo@nus.edu.sg

## Abstract

This paper investigates the dynamical properties of tokens in pre-trained Transformer models and explores their application to improving Transformers. To this end, we analyze the dynamical system governing the continuous-time limit of the pre-trained model and characterize the asymptotic behavior of its solutions. Specifically, we characterize when tokens move closer to or farther from one another over time, depending on the model parameters. We provide sufficient conditions, based on these parameters, to identify scenarios where tokens either converge to zero or diverge to infinity. Unlike prior works, our conditions are broader in scope and more applicable to real-world models. Furthermore, we investigate how different forms of positional encoding – specifically absolute and rotary – affect these dynamical regimes. Empirical evidence reveals that the convergence scenario adversely impacts model performance. Motivated by these insights, we propose simple refinements to Transformer architectures that mitigate convergence behavior in models with absolute or rotary positional encoding. These findings support theoretical foundations and design principles for improving Transformer models.

## 1 Introduction

Transformers [51] have revolutionized multiple domains, demonstrating remarkable success in natural language processing [10, 8, 2, 4, 41], computer vision [13, 43, 38, 32], machine learning [48, 23, 29], and reinforcement learning [22, 5]. Their widespread adoption is largely driven by their ability to leverage large-scale pretraining, enabling efficient knowledge transfer to downstream tasks [40, 12, 39]. Unlike traditional architectures that rely on recurrence or convolution, Transformers are built upon a self-attention mechanism. This mechanism dynamically computes relationships among all tokens in an input sequence, assigning importance scores that dictate how strongly each token influences the others. As a result, self-attention effectively captures intricate dependencies across long

---

[*]Co-first authors.
[†]Co-last authors. Please correspond to tanmn@nus.edu.sg and thieuvo@nus.edu.sg.

39th Conference on Neural Information Processing Systems (NeurIPS 2025).

sequences, making it highly proficient in contextual representation learning [30, 53, 7, 49, 36, 37]. By modeling global interactions across an input, Transformers have surpassed earlier deep learning architectures, solidifying their position as the dominant paradigm in modern artificial intelligence.

Only a limited number of studies in the literature provide a theoretical understanding of the internal structure of learned representations in pre-trained Transformer models (see Section A for a comprehensive list of relevant works). Notably, in the seminal papers [17, 16], the authors modeled Transformers as interacting particle systems and demonstrated that tokens tend to cluster around specific limiting objects determined by the initial tokens, thereby confirming the context-awareness of representations learned by Transformers. It was also observed in [15] that, in the self-attention dynamic, although tokens collapse to a single cluster in infinite time, they remain trapped near a configuration of several clusters for an exponentially long period of time. The authors in [3] analyzed a pure-attention hardmax Transformer model in a similar manner under a discrete framework. A key limitation of these works lies in their reliance on unrealistic and impractical assumptions on model parameters, primarily introduced to support the development of richer theoretical results and technical proofs. Moreover, by focusing exclusively on theoretical analysis, these studies have not yet offered practical applications or insights for improving model performance.

A comprehensive list of related works can be found in Appendix A.

## 1.1 Our Contribution

In this paper, we investigate the internal dynamics of tokens in self-attention mechanisms, together with the effects of absolute and rotary positional encoding, under more realistic and practical assumptions on model parameters, extending prior theoretical studies that often rely on restrictive conditions. We focus on the continuous-time limit of pre-trained Transformer models and systematically analyze the asymptotic behavior of token trajectories – such as convergence, divergence, and token distances – over time. Our contribution is fourth-fold:

1. We provide *conditions that predict whether tokens in self-attention converge to zero or diverge to infinity*. The conditions are more general than those considered in prior works (e.g., [17, 16]) and the theoretical results are validated on pre-trained Transformer models.

2. We show that absolute positional encoding has little effect on token dynamics, while *rotary encoding significantly alters them to promote the divergence of tokens*.

3. We empirically verify that the convergence of tokens to zero negatively impacts model performance. In contrast, in divergence scenarios, tokens organize into a small number of groups, with tokens within the same group diverging to infinity in a consistent direction. This is often beneficial for model performance.

4. Building on these findings, we propose simple yet effective improvements to mitigate convergence scenarios in Transformers with absolute or rotary positional encodings.

To verify our findings, we conducted language modeling experiments on WikiText-103 [34] and EnWik8 [20], and object recognition on ImageNet-1K [11]. The results support our theoretical claim and provide insights to improve Transformer models.

**Organization.** The paper is organized as follows. Section 2 introduces the continuous-time dynamical system representing a pre-trained Transformer. Section 3 presents our main theoretical and empirical results on token dynamics in self-attention. Section 4 extends the analysis to absolute and rotary positional encodings. Section 5 provides empirical validation and proposes improvements to positional encoding schemes. Proofs and additional experiments are provided in the Appendix.

**Notations.** Through this paper, $|| \cdot ||$ denotes the Euclidean norm of a vector. For each subset $A \subseteq \mathbb{R}^D$, we denote by $\mathbf{conv}(A)$ the convex hull of $A$, which is the smallest convex set containing $A$ in $\mathbb{R}^D$. For a square matrix $B$, we denote by $\mathbf{q}_B$ the quadratic form associated to $B$ and by $B_{\text{sym}} = \frac{1}{2}(B + B^\top)$ the symmetric part of $B$. When $B$ is positive definite, we write $B \succ 0$. In case $B$ is negative definite, we write $B \prec 0$.

## 2 Background: Continuous-time Limit of Attention

In this section, we present the dynamical system that represents the continuous-time counterpart of a pre-trained Transformer model.

## 2.1 Continuous-time Limit of a Deep Neural Network

We build on prior work in the literature that explores the dynamical systems underlying the continuous-time limits of deep neural networks (DNNs). A prominent example of DNNs is residual neural network (ResNet). Each layer in a ResNet is a residual block that transforms an input vector $x \in \mathbb{R}^D$ into an output vector $z = x + y(x, \theta) \in \mathbb{R}^D$, where $y = y(x, \theta)$ is a two-layer feed-forward neural network parameterized by $\theta$. The continuous-time counterpart of ResNet is represented as a flow map that takes an input vector $x(0) \in \mathbb{R}^D$ and produces an output vector $x(T) \in \mathbb{R}^D$, governed by the associated dynamical system:

$$x'(t) = y(x(t), \theta(t)), \quad t \in (0, T).$$

There is a substantial body of research examining the interpolation, approximation, and controllability properties of such DNN architectures [28, 54, 27, 47, 44, 6].

## 2.2 Self-Attention Dynamics

In contrast to ResNet, Transformer operates on a sequence of $D$-dimensional tokens rather than solely on individual inputs. Central to the Transformer architecture is the self-attention map. Let $X = [x_1 \ \ldots \ x_L]^\top \in \mathbb{R}^{L \times D}$ be an input sequence of $L$ tokens with $D$ features. Each token is a (column) vector $x_i \in \mathbb{R}^D$. The self-attention map transforms $X$ into the output sequence $Y = [y_1 \ \ldots \ y_L]^\top \in \mathbb{R}^{L \times D_v}$ defined as

$$Y = \text{softmax}\left( \frac{(X \cdot Q) \cdot (X \cdot K)^\top}{\sqrt{D_k}} \right) \cdot X \cdot V. \tag{1}$$

The matrices $Q, K \in \mathbb{R}^{D \times D_k}$ and $V \in \mathbb{R}^{D \times D_v}$ are learnable parameters, and they are called the query, key and value matrices. In our context, we will always assume that $D_v = D$, thus $V$ is a square matrix. We can rewrite equation (1) as

$$y_l = V^\top \cdot \sum_{i=1}^{L} \left( \frac{e^{x_l^\top \cdot W \cdot x_i}}{\sum_{j=1}^{L} e^{x_l^\top \cdot W \cdot x_j}} \right) \cdot x_i,$$

for $l = 1, \ldots, L$, where $W := \frac{1}{\sqrt{D_k}} Q \cdot K^\top$, which is a matrix in $\mathbb{R}^{D \times D}$.

To better understand the internal learning representations of a pre-trained Transformer model, we consider the differential system that governs the continuous-time dynamics of self-attention:

$$\frac{dx_l(t)}{dt} = V^\top \cdot \sum_{i=1}^{L} \left( \frac{e^{x_l(t)^\top \cdot W \cdot x_i(t)}}{\sum_{j=1}^{L} e^{x_l(t)^\top \cdot W \cdot x_j(t)}} \right) \cdot x_i(t), \tag{2}$$

for $l = 1, \ldots, L$, with the initial conditions $(x_1(0), \ldots, x_L(0)) = (x_{10}, \ldots, x_{L0}) \in (\mathbb{R}^D)^L$. The dynamical system for the self-attention with positional encoding will be discussed later in Section 4.

Building on the methodologies introduced in [17, 3], we focus our analysis specifically on the self-attention mechanism - a core component of Transformer architectures - and the role of skip connections within the associated dynamical system. To facilitate tractable analysis, we omit other token-wise operations such as layer normalization and feed-forward networks, and we assume time-invariant model parameters $Q$, $K$, and $V$. While this assumption is primarily for analytical simplicity, it also aligns with parameter-sharing strategies used in models like ALBERT [26] to reduce training costs.

**Remark 2.1** (Scope and Generality). Although our theoretical setup adopts simplifying assumptions, this is standard practice in theoretical work to make the analysis feasible while preserving the ability to capture core empirical behaviors. Several influential studies follow similar simplifications [17, 16, 15, 3]. Notably, our framework generalizes key aspects of [17, 3] by relaxing certain assumptions. Our results are also validated against pre-trained Transformer models, confirming that the theoretical insights carry over to real-world settings.

## 3 Dynamical Properties of Tokens in Self-Attention

We present our main theoretical and empirical results on the dynamical properties of tokens in self-attention dynamics in this section.

**Quadratic Space.** To analyze the dynamical properties of tokens in a more general context, we conduct our study within the framework of quadratic spaces. Each matrix $B \in \mathbb{R}^{D \times D}$ is associated with a quadratic form $\mathbf{q}_B \colon \mathbb{R}^D \to \mathbb{R}$, defined as $\mathbf{q}_B(u) = u^\top \cdot B \cdot u$ for each $u \in \mathbb{R}^D$. The pair $(\mathbb{R}^D, \mathbf{q}_B)$ is referred to as a quadratic space. The quadratic form $\mathbf{q}_B$ is uniquely determined by the symmetric component $B_{\mathrm{sym}} = \frac{1}{2}(B^\top + B)$ of $B$. This means that, for arbitrary matrices $B$ and $B'$, we have $\mathbf{q}_B = \mathbf{q}_{B'}$ if and only if $B_{sym} = B'_{sym}$. In the special case where $\mathbf{q}_B$ is positive definite (i.e., $B_{\mathrm{sym}} \succ 0$), $\mathbf{q}_B$ corresponds to the square of a norm. Conversely, when $\mathbf{q}_B$ is negative definite (i.e., $B_{\mathrm{sym}} \prec 0$), $\mathbf{q}_B$ corresponds to the negative of a squared norm. In particular, in the case $B_{\mathrm{sym}}$ is a definite matrix, the map

$$|| \cdot ||_B = \begin{cases} \sqrt{\mathbf{q}_B(\cdot)}, & \text{if } B_{\mathrm{sym}} \succ 0, \\ \sqrt{-\mathbf{q}_B(\cdot)}, & \text{if } B_{\mathrm{sym}} \prec 0, \end{cases}$$

is actually a norm on $\mathbb{R}^D$, and it is equivariant to the standard Euclidean norm $|| \cdot ||$. Therefore, the dynamical properties of tokens as elements of the quadratic space $(\mathbb{R}^D, \mathbf{q}_B)$ faithfully reflect their dynamical behavior in the standard Euclidean space $(\mathbb{R}^D, || \cdot ||)$.

**Dynamical Properties of Tokens.** To start the analysis of the dynamical properties of tokens, we assume that $(x_1(t), \ldots, x_L(t)) \in C([0, +\infty))^L$ is the unique solution of the dynamical system (2). The existence and uniqueness as well as the well-posedness of this solution was proved in [17, Proposition 6.2]. We consider the function $f \colon \mathbb{R} \times \mathbb{R}^D \to \mathbb{R}$ defined by

$$f(t, u) = \log \left( \sum_{j=1}^{L} e^{u^\top \cdot W \cdot x_j(t)} \right), \quad u \in \mathbb{R}^D, \tag{3}$$

Then the dynamical system (2) can be written as

$$A \cdot \frac{d}{dt} x_l(t) = \frac{\partial}{\partial u} f(t, x_l(t)), \tag{4}$$

where $A = W \cdot (V^\top)^{-1}$, provided that $V$ is invertible. Based on this observation, we characterize the dynamical properties of tokens via $A$ and $W$. In particular, we will show that:

1. **Distances between Tokens.** If $A \prec 0$, then all tokens will move closer to each other as time $t$ approaches infinity. In contrast, if $A \succ 0$, the tokens will either maintain constant distances or move farther away from each other as time $t$ approaches infinity (see Theorem 3.1).

2. **Convergence Scenario.** If $A \prec 0$ and $W \succ 0$, then all tokens will tend to zero as the time $t$ tends to infinity (see Theorem 3.4). Furthermore, our simulations with randomly selected model parameters suggest that the convergence scenario occurs whenever $A_{\mathrm{sym}} \prec 0$ and $W_{\mathrm{sym}} \succ 0$.

3. **Divergence Scenario.** Our simulations with randomly selected model parameters suggest that the divergence scenario occurs whenever the condition "$A_{\mathrm{sym}} \prec 0$ and $W_{\mathrm{sym}} \succ 0$" is violated. In the special case when $V$ has at least one positive eigenvalue and $W$ is arbitrary, we prove that all tokens will diverge to infinity under certain assumption on the initial data (see Theorem 3.6).

We will theoretically prove and empirically validate these observations in the subsequent subsections. The choice of parameters used in the simulation is described in Appendix D.1.

## 3.1 Distances between Tokens

We characterize the dynamical properties of the distances between tokens using the quadratic forms $\mathbf{q}_A$ associated to the matrix $A = W \cdot (V^\top)^{-1}$. In particular, we prove that:

**Theorem 3.1** (Distances between Tokens). *If $A$ is symmetric, then the map $t \mapsto \mathbf{q}_A(x_i(t) - x_j(t))$ is non-decreasing on $[0, +\infty)$. As a consequence,*

(a) *if $A \succ 0$, then $||x_i(t) - x_j(t)||_A$ is non-decreasing on $[0, +\infty)$;*

(b) *if $A \prec 0$, then $||x_i(t) - x_j(t)||_A$ is non-increasing on $[0, +\infty)$.*

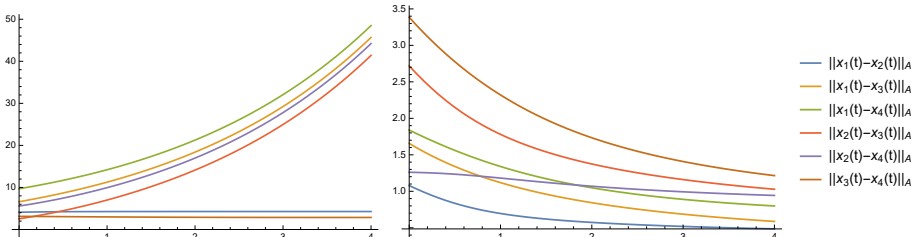

Figure 1: Distances between tokens over time. **Left:** When $A_{\text{sym}} \succ 0$, the distances between tokens do not decrease. **Right:** Conversely, when $A_{\text{sym}} \prec 0$, the distances between tokens do not increase.

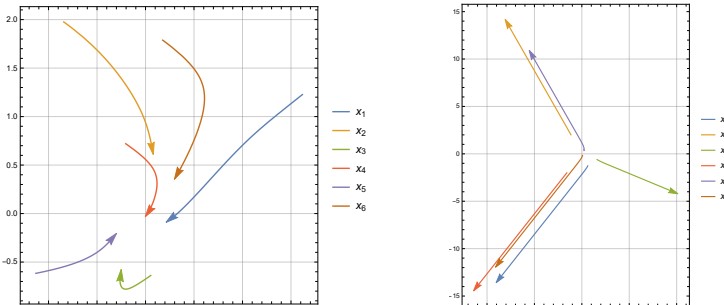

Figure 2: Token trajectories under two configurations. **Left:** When $A_{\text{sym}}$ is negative definite and $W_{\text{sym}}$ is positive definite, all tokens converge to zero as time $t \to \infty$. **Right:** When $A = 2W$, tokens diverge to infinity over time, forming a few distinct groups that move in aligned directions.

Intuitively, Theorem 3.1 states that the tokens will move closer to each other when $A$ is negative definite, while the tokens will either maintain constant distances or move farther away from each other when $A$ is positive definite. This result can be intuitively understood from equation (4). Indeed, one can verify that $f$ is a convex function in $u$. Therefore, the subtraction $x_i(t) - x_j(t)$ will have the same orientation as $\frac{\partial}{\partial u} f(t, x_i(t)) - \frac{\partial}{\partial u} f(t, x_j(t))$, which is equal to the derivative $\frac{d}{dt} A \cdot (x_i(t) - x_j(t))$. As a consequence, the quantity $\frac{d}{dt} \mathbf{q}_A(x_i(t) - x_j(t))$ is always non-negative. The formal proof of this theorem can be found in Appendix B.1.

**Remark 3.2** (Distances between Tokens). In case $A$ is not necessarily symmetric, we observe from randomly chosen of model parameters that the distance $||x_i(t) - x_j(t)||_A$ is non-decreasing (respectively, non-increasing) whenever $A_{\text{sym}} \succ 0$ (respectively, $A_{\text{sym}} \prec 0$).

Figure 1 illustrates the distances between tokens when $A_{\text{sym}}$ is positive (on the left side) and when $A_{\text{sym}}$ is negative (on the right side). As shown in Figure 1, token distances remain constant or increase when $A_{\text{sym}} \succ 0$, and remain constant or decrease when $A_{\text{sym}} \prec 0$, confirming Theorem 3.1.

### 3.2 Convergence Scenario

Our simulations with randomly selected model parameters suggest that the convergence scenario occurs whenever $A_{\text{sym}} \prec 0$ and $W_{\text{sym}} \succ 0$. In this section, we actually prove that when $A \prec 0$ and $W \succ 0$, then all tokens will tend to zero as $t$ approaches infinity. First, we established a relation between the quadratic forms $\mathbf{q}_A$ and $\mathbf{q}_W$ in general setting below:

**Proposition 3.3.** *Assume that $V$ is an invertible matrix and $A = W \cdot (V^\top)^{-1}$ is symmetric. Then,*

$$\frac{d}{dt} \mathbf{q}_A(x_l(t)) \geq 2 - \frac{2L}{e^{\mathbf{q}_W(x_l(t))}},$$

*for all $l = 1, \ldots, L$ and $t \in [0, +\infty)$. As a consequence, if $A \prec 0$ and $W_{\text{sym}} \succ 0$, then all tokens are bounded, i.e there exists $c > 0$ such that $||x_l(t)||_A < c$ for all $t \in [0, +\infty)$ and $l = 1, \ldots, L$.*

In the case $A \prec 0$ and $W \succ 0$, we prove a stronger result as stated in the following theorem:

**Theorem 3.4** (Convergence Scenario). *If $A \prec 0$ and $W \succ 0$, then all tokens converge to zero, i.e. $\lim_{t \to +\infty} x_l(t) = 0$ for all $l = 1, \ldots, L$.*

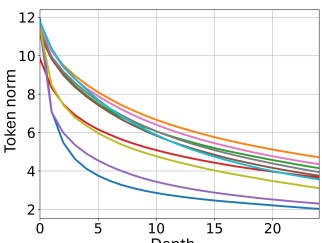
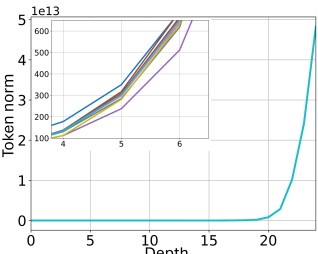
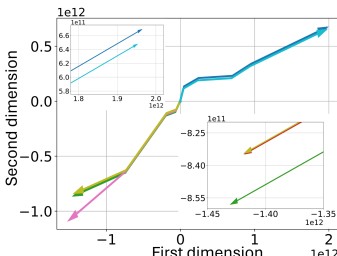

(a) Token norms decrease under the convergence scenario ($A_{\mathrm{sym}} \prec 0$ and $W_{\mathrm{sym}} \succ 0$ ).

(b) Token norms diverge under the divergence scenario ($A_{\mathrm{sym}} \succ 0$ and $W_{\mathrm{sym}} \succ 0$).

(c) Tokens form distinct groups with similar trajectory shapes in divergence scenario.

Figure 3: Token dynamic in pre-trained model when omitting LayerNorm and feed-forward network

The special case where $A = -I_D$ and $W = I_D$ was already proved in [17, Section 8.2]. We generalize their results to a more practical setting which requires weaker assumptions on the model parameters. The proofs of Proposition 3.3 and Theorem 3.4 can be found in Appendix B.2.

**Remark 3.5** (Convergence Scenario). We observe from randomly chosen of model parameters that when $A_{\mathrm{sym}} \prec 0$ and $W_{\mathrm{sym}} \succ 0$, all tokens will converge to zero. Otherwise, the tokens are likely to divergent to infinity.

Figure 2a depicts the trajectories of tokens for the scenario analyzed in Theorem 3.4. Under these conditions, all tokens converge to zero as $t \to \infty$, thus confirming Theorem 3.4.

### 3.3 Divergence Scenario

Our simulation on random selections of model parameters suggests that the divergence scenario is likely to occur whenever the assumption "$A_{\mathrm{sym}} \prec 0$ and $W_{\mathrm{sym}} \succ 0$" in Remark 3.5 is violated. In this subsection, we prove that in the special case where the value matrix $V$ has at least one positive eigenvalue, the divergence scenario indeed occurs under certain assumptions on the initial data.

For each nonzero vector $\mathbf{n} \in \mathbb{R}^D$, we denote by $H_{\mathbf{n}}$ the closed half-space of $\mathbb{R}^D$ with normal vector $\mathbf{n}$ such that zero belongs to its boundary $\partial H_{\mathbf{n}}$. In this case, $\partial H_{\mathbf{n}}$ represents the hyperplane containing zero with normal vector $\mathbf{n}$. In the following, we project the tokens onto the affine line along the eigenvector corresponding to a positive eigenvalue of $V$ to see when token tend to infinity.

**Theorem 3.6** (Divergence Scenario). *Assume that the value matrix $V$ has at least one positive eigenvalue. Let $\mathbf{n}$ be an eigenvector of $V$ corresponding to a positive eigenvalue. Then*

$$\min_{1 \leq i \leq L} \left( \mathbf{n}^\top x_{i0} \right) \leq \mathbf{n}^\top e^{-tV^\top} x_l(t) \leq \max_{1 \leq i \leq L} \left( \mathbf{n}^\top x_{i0} \right),$$

*for all $t \in [0, +\infty)$ and $l = 1, \ldots, L$.*

*As a consequence, if the initial points $x_{10}, \ldots, x_{L0}$ are all in one side of the hyperplane $\partial H_{\mathbf{n}}$, and if $V$ has only positive eigenvalues, then $\lim_{t \to +\infty} \|x_l(t)\| = +\infty$ for all $l$.*

The proofs of Theorem 3.6 can be found in Appendix B.3. Figure 2b illustrates the divergence scenario with $A = 2W$ (thus $V = 2I_D$). We further empirically verify that the divergence-scenario constraints arise in practical pretrained models; see Appendix D.6.2.

### 3.4 Tokens' Dynamic in Practical Transformers Architecture

This section presents an empirical study of token dynamics in a practical Transformer architecture. We analyze a 24-layer model pre-trained on WikiText-103 [34], enforcing our theoretical framework by constraining the symmetric matrices $W_{\mathrm{sym}}$ and $A_{\mathrm{sym}}$ to be either positive definite or negative definite (see Appendix D.2 for details).

We begin by empirically verifying the convergence–divergence behavior of simplified model variants that omit feed-forward networks and LayerNorms, directly validating our theoretical results. When $A_{\mathrm{sym}} \prec 0$ and $W_{\mathrm{sym}} \succ 0$, token norms contract monotonically across layers, potentially collapsing toward zero if number of layers increased (Figure 3a). By contrast, under $A_{\mathrm{sym}} \succ 0$ and $W_{\mathrm{sym}} \succ 0$, norms grow without bound (Figure 3b), consistent with Proposition 3.3. Replicating these experiments

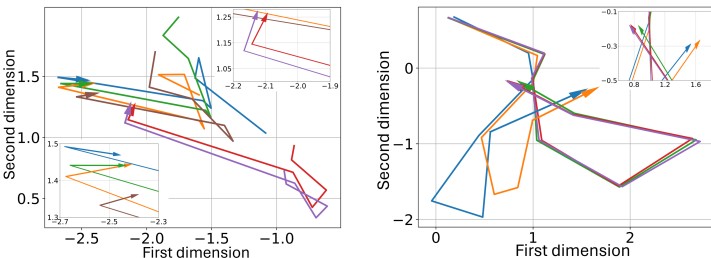

Figure 4: Tokens' trajectory in the later layers of a Transformer forms clusters with similar shapes in model (left) with LayerNorm, no feedforward, and (right) with both LayerNorm and feedforward.

on the GPT-2 pre-trained model [40]–with LayerNorms and feed-forward networks removed–exhibits the same divergent behavior (see Appendix D.6.3).

We then demonstrate clustering behavior in both simplified and full models. With $A_{\text{sym}} \succ 0$ and $W_{\text{sym}} \succ 0$, projecting token trajectories into two dimensions reveals groups with aligned directional patterns (Figure 3c). Even in the full configuration with FFNs and LayerNorms, the model maintains persistent directional clustering in deeper layers (Figure 4). Finally, we observe similar clustering behavior on GPT-2 without enforcing any constraints (see Appendix D.5.1).

## 4 Beyond Self-Attention Dynamic: Effect of Positional Encoding

In practice, Transformer models typically incorporate positional encodings into self-attention mechanisms to enhance expressivity and computational efficiency. In this section, we analyze how positional encodings influence token dynamics. Among common approaches, absolute and rotary positional encodings are predominant. We demonstrate that, under rotary positional encoding, tokens are more likely to exhibit divergent behavior rather than convergent dynamics in comparison with the self-attention with/without absolute positional encoding.

### 4.1 Absolute Positional Encoding.

The self-attention map with absolute positional encoding [51, 50] transforms the input sequence $X$ into the output sequence $Y$ as

$$y_l = V^\top \cdot \sum_{i=1}^{L} \left( \frac{e^{(x_l+p_l)^\top \cdot W \cdot (x_i+p_i)}}{\sum_{j=1}^{L} e^{(x_l+p_l)^\top \cdot W \cdot (x_j+p_j)}} \right) \cdot (x_i + p_i),$$

for $l = 1, \ldots, L$, where $p_i = [p_{i,1}, \ldots, p_{i,D}]^\top \in \mathbb{R}^D$. A common choice is to either learn the positional embeddings $p_i$ jointly with the model parameters, or to fix them using a sinusoidal scheme:

$$p_{i,j} = \begin{cases} \sin(i \cdot 10000^{-\frac{j}{D}}), & \text{if } j \text{ is even,} \\ \cos(i \cdot 10000^{-\frac{j-1}{D}}), & \text{if } j \text{ is odd.} \end{cases}$$

The differential system that governs the continuous-time dynamics of the self-attention with absolute positional encoding is given by

$$\frac{dx_l(t)}{dt} = V^\top \cdot \sum_{i=1}^{L} \left( \frac{e^{(x_l(t)+p_l)^\top \cdot W \cdot (x_i(t)+p_i)}}{\sum_{j=1}^{L} e^{(x_l(t)+p_l)^\top \cdot W \cdot (x_j(t)+p_j)}} \right) \cdot (x_i(t) + p_i), \tag{5}$$

for $l = 1, \ldots, L$.

**Remark 4.1** (Absolute positional encoding has minimal impact). The dynamical properties of the self-attention with and without absolute are similar, since the above differential system can be transformed into the original self-attention dynamic (2) by the transition $x_l \mapsto x_l + p_l$. In particular, it follows from Theorem 3.4 that: when $A \prec 0$ and $W \succ 0$, the solution $X(t)$ of equation (5) will converge to the sequence $P = (-p_1, \ldots, -p_L)$. In addition, it follows from Theorem 3.6 that, in case $V$ has at least one positive eigenvalue, tokens will diverge to infinite under certain assumptions on the initial conditions.

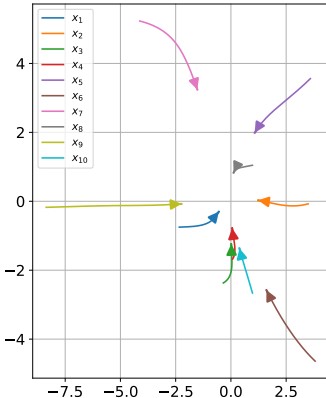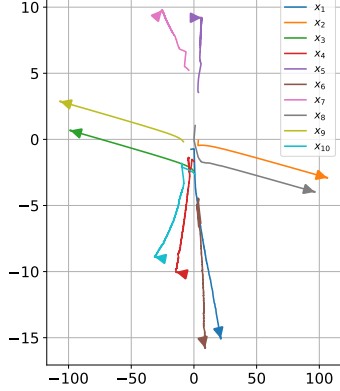

Figure 5: By introducing the additional term $\Delta W_{li}$, the system's behavior shifts from a convergent regime (left) to a divergent regime (right).

## 4.2 Rotary Positional Encoding.

In contrast to absolute positional encoding, for even dimension $D$, the rotary positional encoding [31, 45] maps the input $X$ into the output $Y$ as

$$y_l = V^\top \cdot \sum_{i=1}^{L} \left( \frac{e^{x_l^\top \cdot W_{li} \cdot x_i}}{\sum_{j=1}^{L} e^{x_l^\top \cdot W_{lj} \cdot x_j}} \right) \cdot x_i, \tag{6}$$

for $l = 1, \ldots, L$, where

$$W_{li} = \frac{1}{\sqrt{D_k}} \left( Q \cdot K^\top + \overline{Q} \cdot R_{\theta, i-l}^D \cdot \overline{K}^\top \right) \in \mathbb{R}^{D \times D}, \tag{7}$$

with $\overline{Q}, \overline{K}$ are two additional learnable matrices and

$$R_{\Theta, m}^D = \text{blockdiag} \left( \begin{pmatrix} \cos m\theta_1 & -\sin m\theta_1 \\ \sin m\theta_1 & \cos m\theta_1 \end{pmatrix}, \ldots, \begin{pmatrix} \cos m\theta_{D/2} & -\sin m\theta_{D/2} \\ \sin m\theta_{D/2} & \cos m\theta_{D/2} \end{pmatrix} \right)$$

where $\Theta = \left\{ \theta_i = 10000^{-2(i-1)/D}, \quad i = 1, 2, \ldots, D/2 \right\}$.

The continuous-time dynamics of the self-attention with rotary positional encoding can be described via the differential system

$$\frac{dx_l(t)}{dt} = V^\top \cdot \sum_{i=1}^{L} \left( \frac{e^{x_l(t)^\top \cdot W_{li} \cdot x_i(t)}}{\sum_{j=1}^{L} e^{x_l(t)^\top \cdot W_{lj} \cdot x_j(t)}} \right) \cdot x_i(t), \tag{8}$$

for $l = 1, \ldots, L$. Rotary positional encoding is an essential component of latent attention which is at the core of Deepseek [31].

**Remark 4.2** (Rotary positional encoding encourages token divergence). Unlike absolute positional encoding, the rotary positional encoding exhibits markedly different behavior as its encoding encourages token divergence. Indeed, the additional term $\Delta W_{li} = \overline{Q} \cdot R_{\theta, i-l}^D \cdot \overline{K}^\top$ in the query-key interaction matrix $W_{li}$, as defined in equation (7), can inhibit the system from entering the convergence regime, as shown in Figure 5. As a result, in self-attention mechanisms with rotary positional encoding, the divergence scenario tends to occur more frequently compared to those employing absolute or no positional encoding.

To support our findings, we visualize the evolution of token norms and pairwise L2 distances in a pre-trained Transformer model without $\text{LayerNorm}$ and feed-forward layers, comparing Rotary Positional Encodings (RoPE) with sinusoidal positional encodings. Figure 6 shows that both token norms and token distances diverge faster with RoPE than with sinusoidal encodings, suggesting that RoPE mitigates the convergence aspect in self-attention.

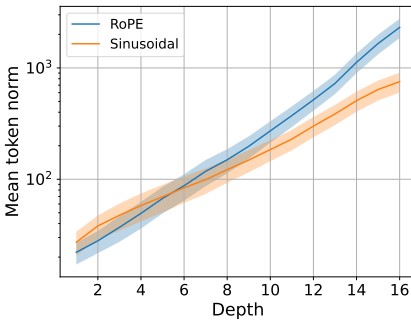 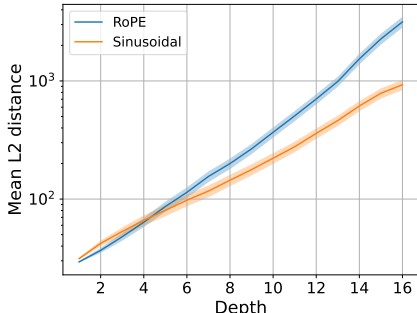

Figure 6: Token norm and distance throughout layers of a pre-trained model with RoPE and Sinusoidal positional encoding.

Table 1: Performance of Transformers with sinusoidal positional encoding across scenarios on language modeling task.

| Scenario | EnWik8 BPC ($\downarrow$) | WikiText-103 PPL ($\downarrow$) Valid | Test |
|---|---|---|---|
| *Baseline* | 1.331 | 31.63 | 32.37 |
| Convergence | 1.350 | 32.24 | 33.05 |
| Intermediate | 1.345 | 31.91 | 32.78 |
| Divergence | **1.324** | **31.12** | **32.07** |

Table 2: Top 1 and Top 5 Validation Accuracy of DeiT (learnable positional encoding) across different scenarios on ImageNet1K.

| Scenario | Top 1 Acc ($\uparrow$) | Top 5 Acc ($\uparrow$) |
|---|---|---|
| *DeiT Baseline* | 71.80 | 91.01 |
| Convergence | 71.34 | 90.64 |
| Intermediate | 71.60 | 90.91 |
| Divergence | **71.96** | **91.05** |

## 5 Experiments

In this section, we provide empirical validation of our theoretical findings and introduce enhancements to absolute positional encoding and Rotary Positional Embedding (RoPE) in practical Transformer models. We conduct experiments on three benchmark tasks: language modeling on WikiText-103 [34] and EnWik8 [20], and image classification on ImageNet-1K [11]. Our goals are twofold: (1) to show that convergence behavior adversely affects the performance of Transformers and should thus be mitigated; and (2) to demonstrate that encouraging divergence in Transformers improves performance. Throughout the experiments, we compare our modified Transformers with the baseline Transformers of the same configuration. Our results are averaged over 5 runs. Detailed information about the model architecture, hyperparameters, and training procedures is provided in Appendix D.

### 5.1 Negative Effects of the Convergence Scenario

We empirically examine the impact of convergence and divergence dynamics predicted by our analysis across different Transformer architectures. Specifically, we evaluate models on the WikiText-103 and EnWik8 language modeling tasks using sinusoidal positional encoding, and on ImageNet-1K using a Vision Transformer variant DeiT [50] which uses learnable positional encoding. The parameters in attention layers are explicitly constrained to conform to these distinct scenarios. In the convergence scenario, we impose the conditions $W_{\text{sym}} \succ 0$ and $A_{\text{sym}} \prec 0$. Conversely, in the divergence scenario, we require $W_{\text{sym}} \succ 0$ and $A_{\text{sym}} \succ 0$. We additionally examine an intermediate scenario in which $W_{\text{sym}} \succ 0$ and $A_{\text{sym}}$ contains an equal number of positive and negative eigenvalues. Further details on the benchmark setup and the implementation of these constraints are provided in Appendix D.3 and Appendix D.2, respectively.

Table 1 reports model performance across the baseline and three constrained settings. As the constraints shift toward convergence, performance consistently deteriorates. In contrast, the divergence setting achieves the best results, yielding the lowest BPC and perplexity. On WikiText-103, the divergence case outperforms the convergence and baseline settings by reducing validation perplexity by **1.12** and **0.51**, respectively. On EnWik8, it lowers test BPC to 1.324, compared to 1.350 under convergence and 1.331 under the baseline. These results suggest that divergence constraints improve model performance, while convergence constraints have a detrimental effect.

Table 3: Bits Per Characters (BPC) and Perplexity (PPL) of Transformers with Rotary Positional Encoding across different scenarios on EnWik8 and WikiText-103 language modeling.

| Scenario | EnWik8 Pretrain | WikiText-103 Pretrain | |
|---|---|---|---|
| | Test BPC ($\downarrow$) | Valid PPL ($\downarrow$) | Test PPL ($\downarrow$) |
| *Transformer + RoPE* | 1.295 | 31.37 | 32.35 |
| Transformer + RoPE + $\lambda I_D$ | 1.288 | 31.10 | 32.09 |
| Transformer + RoPE + $\lambda A$ | **1.281** | **31.06** | **32.04** |

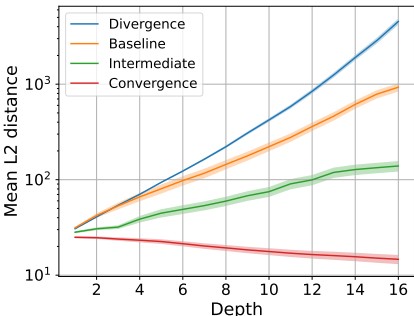 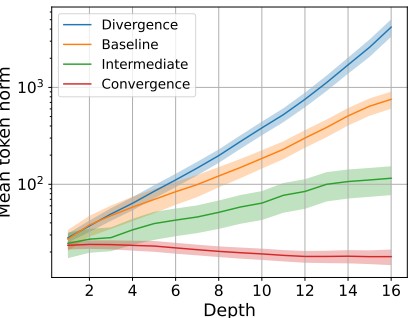

Figure 7: Mean token distance (left) and mean token norm (right) across Transformer layers in a pre-trained model without LayerNorm or feedforward network.

To analyze the impact of convergence and divergence constraints, we plot the mean token norm and mean pairwise L2 distance across layers (without LayerNorm or feed-forward). As shown in Figure 7, both metrics decrease under the convergence constraint, whereas they rise under the intermediate and divergence regimes, with divergence exhibiting the steepest increase. These empirical trends validate our theoretical analysis in Section 3 and suggest that the intermediate setting combines aspects of both convergence and divergence regimes.

## 5.2 Promoting Divergence in Rotary Positional Embedding

Motivated by the influence of the additional term $\Delta W_{li}$ on the query-key interaction matrix $W_{li}$ in promoting divergent token dynamics under RoPE, we propose a simple yet effective modification to further mitigate convergence - an undesirable regime shown to negatively affect model performance. Specifically, we add a learnable regularization term $\lambda I_D$ or $\lambda A \in \mathbb{R}^{D \times D}$ to $W_{li}$, where $\lambda$ is a learnable negative scalar, $I_D$ is the identity matrix, and $A$ is a learnable diagonal matrix with strictly positive entries. This modification effectively learns to subtract a positive quantity from the diagonal of $W_{li}$, promoting negative eigenvalues, thus discouraging convergence scenario in attention layers.

We provide empirical evidence on WikiText-103 and EnWik8 demonstrating our findings' practical relevance. As shown in Table 3, even minimal intervention yields measurable gains. On EnWik8, adding $\lambda I_D$ reduces Test BPC from 1.295 to 1.288, and $\lambda A$ further reduces it to 1.281. On WikiText-103, both variants outperform RoPE, with $\lambda A$ achieving a 0.3-point improvement in Validation and Test perplexity. These results, while modest, are consistent with our theoretical predictions and validate the utility of encouraging divergence in self-attention dynamics.

## 6 Conclusion

We analyzed token dynamics in self-attention and leveraged these insights to improve Transformer performance. Through theoretical investigation, we derived conditions under which tokens converge or diverge during self-attention and explored how positional encodings - such as absolute and rotary - affect these dynamics. Empirical evaluations on pre-trained Transformers supported our findings, highlighting the negative impact of token convergence on performance. To address this, we proposed simple enhancements to self-attention that yielded measurable improvements. A limitation of our analysis is its focus on self-attention, omitting components like layer normalization and feed-forward networks. However, empirical validation confirms our conclusions remain relevant in practice. Extending our framework to the full Transformer architecture is a promising direction for future research.

## Acknowledgments and Disclosure of Funding

This research / project is supported by the National Research Foundation Singapore under the AI Singapore Programme (AISG Award No: AISG2-TC-2023-012-SGIL). This research / project is supported by the Ministry of Education, Singapore, under the Academic Research Fund Tier 1 (FY2023) (A-8002040-00-00, A-8002039-00-00). This research / project is also supported by the NUS Presidential Young Professorship Award (A-0009807-01-00) and the NUS Artificial Intelligence Institute–Seed Funding (A-8003062-00-00).

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

# Supplement to "Dynamical Properties of Tokens in Self-Attention and Effects of Positional Encoding"

## Contents

## A  Related Works

**Transformers as Interacting Particle Systems.**   Particle systems offer a novel perspective to understand the dynamics of Transformer models and inspire architectural innovations. In [33], the Transformer architecture is mathematically framed as a numerical ODE solver for a convection-diffusion equation in a multi-particle dynamic system. Similarly, leveraging insights from numerical ODE solvers, [14] introduces TransEvolve, a temporal evolution scheme inspired by interacting particle dynamics. Furthermore, the ODE governing Transformer dynamics is closely connected to the extensive literature on nonlinear systems, including flocking phenomena [19], the Kuramoto model [25, 1], consensus formation [24, 35], opinion formation [21, 42], and systems of self-driven particles [52].

**Clustering Effects of Transformers.**   Research on interacting particle systems has shown that tokens in Transformer models exhibit a long-time clustering phenomenon. Geshkovski et al proved in [17] that tokens cluster around limiting objects determined by their initial values, highlighting their context awareness. This was extended to the study of metastability of self-attention with layer normalization in [16, 15]. Additionally, [3] proved that tokens in pure-attention hardmax Transformer models asymptotically converge to clustered equilibria.

## B  Dynamical Properties of Tokens in Self-Attention

Recall that the dynamical system governing the continuous-time dynamic of the self-attention is given by:

$$\frac{dx_l(t)}{dt} = V^\top \cdot \sum_{i=1}^{L} \left( \frac{e^{x_l(t)^\top \cdot W \cdot x_i(t)}}{\sum_{j=1}^{L} e^{x_l(t)^\top \cdot W \cdot x_j(t)}} \right) \cdot x_i(t), \quad l = 1, \ldots, L, \tag{9}$$

with the initial condition $(x_1(0), \ldots, x_L(0)) = (x_{10}, \ldots, x_{L0}) \in (\mathbb{R}^D)^L$. In this system, $W = Q \cdot K^\top$. The matrices $Q, K, V$ are learnable matrices and they are called queries, keys and values matrices, respectively. In our setting, the parameters $W$ and $V$ are assumed to be time-independent.

Set $A = W \cdot (V^\top)^{-1}$. We will prove the following observations in the subsequent subsections.

> **Distances Between Tokens.** If $A \prec 0$, then all tokens will tend to move closer and closer to each other as time $t$ approaches infinity. In contrast, if $A \succ 0$, the tokens will either tend to maintain constant distances or move farther away from each other as time $t$ approaches infinity (see Theorem B.2).

> **Convergence Scenario.** If $A \prec 0$ and $W \succ 0$, then all tokens will tend to zero as the time $t$ tends to infinity (see Theorem B.6).

> **Divergence Scenario.** If $V$ has at least one positive eigenvalue and $W$ is arbitrary, then all tokens will diverge to infinity under certain assumption on the initial data (see Theorem 3.6).

### B.1  Distances Between Tokens Over Time

In this subsection, we analyze the dynamical properties of the distances between tokens as the time $t$ increases. We start with the following lemma, which can be seen as a refinement of [17, Lemma 7.1].

**Lemma B.1.** *Let $W$ be an arbitrary matrix in $\mathbb{R}^{D \times D}$. For each $x_1, \ldots, x_L$ in $\mathbb{R}^D$, the function $f \colon \mathbb{R}^D \to \mathbb{R}$ defined by*

$$f(u) = \log \left( \sum_{j=1}^{L} e^{u^\top \cdot W \cdot x_j} \right), \quad u \in \mathbb{R}^D,$$

*is a convex function.*

*Proof.* For arbitrary $u, v \in \mathbb{R}^D$, we have

$$e^{f(u)+f(v)} - e^{2f\left(\frac{u+v}{2}\right)} = \left(\sum_{i=1}^{L} e^{u^\top \cdot W \cdot x_i}\right) \cdot \left(\sum_{j=1}^{L} e^{v^\top \cdot W \cdot x_j}\right) - \left(\sum_{k=1}^{L} e^{\left(\frac{u+v}{2}\right)^\top \cdot W \cdot x_k}\right)^2$$

$$= \sum_{i,j} \frac{1}{2} \left(e^{(u+v)^\top \cdot W \cdot x_i} + e^{(u+v)^\top \cdot W \cdot x_j}\right) - \sum_{i,j} e^{\frac{u+v}{2} \cdot W \cdot x_i + \frac{u+v}{2} \cdot W \cdot x_j}$$

$$= \sum_{i,j} \frac{1}{2} \left(e^{\frac{u+v}{2} \cdot W \cdot x_i} - e^{\frac{u+v}{2} \cdot W \cdot x_j}\right)^2$$

$$\geq 0.$$

Therefore, $f(u) + f(v) \geq 2f\left(\frac{u+v}{2}\right)$. Hence, $f$ is convex. $\qquad\square$

The function $f$ defined in the above lemma is not strictly convex as the equality happens when $u + v = 0$.

**Theorem B.2.** *Let $(x_1(t), \ldots, x_L(t)) \in C^\infty([0, +\infty)])^L$ be a solution of the dynamical system* (9). *Assume that $V$ is invertible and $A = W \cdot (V^\top)^{-1}$ is symmetric. Then the map $t \mapsto \mathbf{q}_A(x_i(t) - x_j(t))$ is non-decreasing on $[0, +\infty)$.*

*As a consequence,*

(a) *if $A \succ 0$, then $||x_i(t) - x_j(t)||_A$ is non-decreasing on $[0, +\infty)$;*

(b) *if $A \prec 0$, then $||x_i(t) - x_j(t)||_A$ is non-increasing on $[0, +\infty)$.*

*Proof.* Fix an arbitrary time $s \in (0, +\infty)$. Let $f$ be the function defined in Lemma B.1 (with $x_j$ there is replaced by $x_j(s)$ in this proof). Then we see that

$$\frac{\partial f}{\partial u}(u) = W \cdot \sum_{i=1}^{L} \frac{e^{u^\top \cdot W \cdot x_i(s)}}{\sum_{j=1}^{L} e^{u^\top \cdot W \cdot x_j(s)}} \cdot x_i(s).$$

Therefore, from the dynamical system (9), we have

$$\frac{\partial f}{\partial u}(x_l(s)) = A \cdot \frac{d}{ds} x_l(s), \quad l = 1, \ldots, L. \tag{10}$$

Since $f$ is convex, we have

$$(x_i(s) - x_j(s))^\top \left(\frac{\partial f}{\partial u}(x_i(s)) - \frac{\partial f}{\partial u}(x_j(s))\right) \geq 0.$$

From equation (10), we have

$$(x_i(s) - x_j(s))^\top \cdot A \cdot \left(\frac{d}{ds} x_i(s) - \frac{d}{ds} x_j(s)\right) \geq 0.$$

Since $A$ is symmetric, it follows from the above inequality that:

$$\frac{d}{ds} \mathbf{q}_A(x_i(s) - x_j(s)) \geq 0.$$

This shows that the map $t \mapsto \mathbf{q}_A(x_i(s) - x_j(s))$ is non-decreasing on $[0, +\infty)$. The items $(a)$ and $(b)$ are obtained due to the definition of $|| \cdot ||_A$. $\qquad\square$

## B.2 Convergence Scenario

In this section, we will prove that when $A \prec 0$ and $W \succ 0$, the solution of the dynamical system (9) tends to zero as $t$ approaches infinity. The special case where $A = -I_D$ and $W = I_D$ was already proved in [17, Section 8.2]. We borrow the proof technique from [17, Section 8.2] and carefully refine it so that it applies to a broader generalization.

**Proposition B.3.** *Let* $(x_1(t), \ldots, x_L(t)) \in C^\infty([0, +\infty)])^L$ *be a solution of the dynamical system* (9). *Assume that* $V$ *is invertible and* $A = W \cdot (V^\top)^{-1}$ *is symmetric. Then, we have*

$$\frac{d}{dt} \mathbf{q}_A(x_l(t)) \geq 2 - \frac{2L}{e^{\mathbf{q}_W(x_l(t))}},$$

*for all* $l = 1, \ldots, L$ *and* $t \in [0, +\infty)$. *As a consequence, if* $A \prec 0$ *and* $W_{sym} \succ 0$, *then* $||x_l(t)||_A$ *is bounded for all* $l = 1, \ldots, L$.

*Proof.* Since $A$ is symmetric, we have

$$\frac{1}{2} \frac{d}{dt} \mathbf{q}_A(x_l(t)) = x_l(t)^\top \cdot W \cdot (V^\top)^{-1} \cdot \frac{d}{dt} x_l(t)$$

$$= \frac{\sum_{i=1}^L e^{x_l(t)^\top \cdot W \cdot x_i(t)} \cdot x_l(t)^\top \cdot W \cdot x_i(t)}{\sum_{j=1}^L e^{x_l(t)^\top \cdot W \cdot x_j(t)}}$$

$$\geq \frac{\sum_{i=1}^L e^{x_l(t)^\top \cdot W \cdot x_i(t)} - L}{\sum_{j=1}^L e^{x_l(t)^\top \cdot W \cdot x_j(t)}}$$

$$= 1 - \frac{L}{\sum_{j=1}^L e^{x_l(t)^\top \cdot W \cdot x_j(t)}}.$$

In the above estimation, to obtain the inequality in the third line, we used the fact that $e^\lambda \lambda \geq e^\lambda - 1$ for all $\lambda \in \mathbb{R}$. As a consequence, we have

$$\frac{d}{dt} \mathbf{q}_A(x_l(t)) \geq 2 - \frac{2L}{e^{\mathbf{q}_W(x_l(t))}}, \tag{11}$$

as claimed.

Next, assume that $A \prec 0$ and $W_{\mathrm{sym}} \succ 0$. There is a constant $c > 0$ such that $|| \cdot ||_A^2 \geq c|| \cdot ||_W^2$. Then it follows from equation (11) that

$$-\frac{d}{dt} ||x_l(t)||_A^2 \geq 2 - \frac{2L}{e^{c||x_l(t)||_A^2}},$$

or equivalently,

$$\frac{d}{dt} ||x_l(t)||_A^2 \leq -2 + \frac{2L}{e^{c||x_l(t)||_A^2}},$$

Hence, $||x_l(t)||_A^2 \leq \frac{1}{c} \log \left( e^{-2ct} \left( e^{c||x_l(0)||_A^2} - L \right) + L \right)$, which is bounded. $\qquad \square$

The following lemma studies the limitation of the derivative of tokens in case $A \prec 0$ and $W \succ 0$.

**Lemma B.4.** *Let* $(x_1(t), \ldots, x_L(t)) \in C^\infty([0, +\infty)])^L$ *be the unique solution of the dynamical system* (9). *Set* $A = W \cdot (V^\top)^{-1}$. *If* $A \prec 0$ *and* $W \succ 0$, *then*

$$\int_0^{+\infty} \left\| \frac{d}{ds} x_l(s) \right\|_A^2 ds < +\infty,$$

*for all* $l$. *In particular, we have* $\lim_{t \to +\infty} \frac{d}{dt} x_l(t) = 0$ *for all* $l$.

*Proof.* Consider the function $h \colon [0, +\infty) \to \mathbb{R}$ defined by

$$h(t) = \sum_{i=1}^L \sum_{j=1}^L e^{x_i(t)^\top \cdot W \cdot x_j(t)}.$$

Then $h$ is a positive function. Since $W$ is symmetric, the derivative of $h$ is

$$\frac{d}{dt} h(t) = 2 \sum_{i=1}^L \sum_{j=1}^L e^{x_i(t)^\top \cdot W \cdot x_j(t)} \cdot \frac{d}{dt} x_i(t)^\top \cdot W \cdot x_j(t)$$

$$= 2 \sum_{i=1}^L \frac{d}{dt} x_i(t)^\top \cdot W \cdot \left( \sum_{j=1}^L e^{x_i(t)^\top \cdot W \cdot x_j(t)} \cdot x_j(t) \right)$$

$$= 2 \sum_{i=1}^L \frac{d}{dt} x_i(t)^\top \cdot A \cdot \frac{d}{dt} x_i(t) \cdot \left( \sum_{j=1}^L e^{x_i(t)^\top \cdot W \cdot x_j(t)} \right).$$

Since $A \prec 0$, we have $\frac{d}{dt}x_i(t)^\top \cdot A \cdot \frac{d}{dt}x_i(t) = -\left\| \frac{d}{dt}x_i(t) \right\|_A^2$. Therefore, we can proceed the above expression as

$$\frac{d}{dt}h(t) = -2\sum_{i=1}^{L}\left\| \frac{d}{dt}x_i(t) \right\|_A^2 \cdot \left( \sum_{j=1}^{L} e^{x_i(t)^\top \cdot W \cdot x_j(t)} \right), \tag{12}$$

which is nonpositive. As a consequence, $h(t)$ is non-increasing. Thus, $\lim_{t\to+\infty} h(t)$ exists and finite.

Next, since $A \prec 0$ and $W \succ 0$, it follows from Proposition B.3 that $x_l(t)$ are bounded for all $l = 1, \ldots, L$. Therefore, there exists $\epsilon > 0$ such that

$$\sum_{j=1}^{L} e^{x_i(t)^\top \cdot W \cdot x_j(t)} \geq \epsilon,$$

for all $t \in [0, +\infty)$ and $i = 1, \ldots, L$. It follows from equation (12) that

$$\frac{d}{dt}h(t) \leq -2\epsilon \left\| \frac{d}{dt}x_l(t) \right\|_A^2.$$

By taking the integral both sides, we see that

$$\int_0^{+\infty} \left\| \frac{d}{ds}x_l(s) \right\|_A^2 ds \leq \frac{1}{2\epsilon}(h(0) - \lim_{s\to+\infty} h(s)) < +\infty.$$

The lemma is then proved. $\qquad\square$

We will also require the following lemma, which holds for any matrix $W$ whose symmetric component $W_{\text{sym}}$ is either positive definite or negative definite. The special case where $W = I_D$ was established in [17, Lemma 8.8]. Our proof, however, is simpler and extends to more general matrices $W$.

**Lemma B.5.** *Assume that $W_{\text{sym}}$ is (either positive or negative) definite. Let $x_1^*, \ldots, x_L^*$ be point in $\mathbb{R}^D$ such that*

$$\sum_{j=1}^{L} e^{(x_l^*)^\top \cdot W \cdot x_j^*} \cdot x_j^* = 0, \quad \forall l = 1, \ldots, L.$$

*Then $x_1^* = \ldots = x_L^* = 0$.*

*Proof.* Consider the function $g\colon \mathbb{R}^D \to \mathbb{R}$ defined by

$$g(u) = \sum_{l=1}^{L} e^{u^\top \cdot W \cdot x_l^*}, \quad \forall u \in \mathbb{R}^D.$$

Then for arbitrary $u, v \in \mathbb{R}^D$, we have

$$g(u) + g(v) - 2g\left( \frac{u+v}{2} \right) = \sum_{l=1}^{L} \left( e^{\frac{1}{2}u^\top \cdot W \cdot x_l^*} - e^{\frac{1}{2}v^\top \cdot W \cdot x_l^*} \right)^2 \geq 0.$$

Therefore $g$ is convex. From the hypothesis, we have

$$\nabla g(x_1^*) = \ldots = \nabla g(x_L^*) = 0.$$

This means that $x_1^*, \ldots, x_L^*$ are global minimum of $g$ and the values of $g$ at these points are all equal. Since $g$ is convex, $g$ achieves the global minimal value on the convex hull $\mathbf{conv}(\{x_l^*\}_{l=1}^L)$. As a consequence, we have

$$g(x_i^*) = g(x_j^*) = g\left( \frac{x_i^* + x_j^*}{2} \right),$$

for all $i, j$. Therefore,

$$0 = g(x_i^*) + g(x_j^*) - 2g\left( \frac{x_i^* + x_j^*}{2} \right) = \sum_{l=1}^{L} \left( e^{\frac{1}{2}x_i^\top \cdot W \cdot x_l^*} - e^{\frac{1}{2}x_j^\top \cdot W \cdot x_l^*} \right)^2.$$

This happens only when

$$\frac{1}{2}x_i^\top \cdot W \cdot x_l^* = \frac{1}{2}x_j^\top \cdot W \cdot x_l^*,$$

or equivalently,

$$(x_i - x_j)^\top \cdot W \cdot x_l^* = 0,$$

for all $l = 1, \dots, L$. In particular, we have

$$\mathbf{q}_W(x_i^* - x_j^*) = (x_i^* - x_j^*)^\top \cdot W \cdot x_i^* - (x_i - x_j)^\top \cdot W \cdot x_j^* = 0.$$

Since $W_{\mathrm{sym}}$ is definite, $\mathbf{q}_W$ is nondegenerate and $x_i^* = x_j^*$. Thus $x_1^* = \dots = x_L^*$. The only possibility is $x_1^* = \dots = x_L^* = 0$. $\qquad\square$

We are ready to prove the main result of the convergence scenario.

**Theorem B.6.** *Let $(x_1(t), \dots, x_L(t)) \in C^\infty([0, +\infty))$ be a solution of the dynamical system* (9). *If $A \prec 0$ and $W \succ 0$, then $\lim_{t\to+\infty} x_l(t) = 0$ for all $l = 1, \dots, L$.*

*Proof.* Set $X(t) = (x_1(t), \dots, x_L(t))$. We need to prove that $\lim_{t\to+\infty} X(t) = 0$. Assume that this is not the case. According to item $(a)$ of Proposition B.3, $X(t)$ lies in a compact subspace of $(\mathbb{R}^D)^L$. Therefore, there exists a sequence $\{t_k\}_k$ in $[0, +\infty)$ such that $\lim_{k\to+\infty} t_k = +\infty$ and $\lim_{k\to+\infty} X(t_k) = X^*$ for some $X^* = (x_1^*, \dots, x_L^*) \in (\mathbb{R}^D)^L$ and $X^* \neq 0$. As a consequence, we have $\lim_{k\to+\infty} x_l(t_k) = x_l^*$ for each $l = 1, \dots, L$. While, according to Lemma B.4, $\lim_{k\to+\infty} \frac{d}{dt}x_l(t_k) = 0$. Therefore, from the dynamical system (9), we obtain

$$\sum_{i=1}^{L} \frac{e^{(x_l^*)^\top \cdot W \cdot x_i^*}}{\sum_{j=1}^{L} e^{(x_l^*)^\top \cdot W \cdot x_j^*}} \cdot x_i^* = 0, \quad l = 1, \dots, L. \tag{13}$$

Then it follows from Lemma B.5 that $x_1^* = \dots = x_L^* = 0$. However, this contradict to the fact that $X^* \neq 0$. Hence, $\lim_{t\to+\infty} X(t) = 0$ and the theorem is proved. $\qquad\square$

## B.3 Divergence Scenario

To simplify the technical details, we will consider the case where $A = \lambda W$, i.e. $V = \frac{1}{\lambda}I_D$, for some positive real number $\lambda$. The case where $A$ and $W$ have the same signs but $A \neq \lambda W$ may require certain adaptations. In particular, we will prove that when $V = \lambda I_D$ and $W$ is arbitrary, all tokens tend to infinity at an exponential rate (under certain assumptions on the initial conditions). The case where $V = W = I_D$ was already solved in [17, Section 8]. We borrow the technique from there and modify it to ensure it works for all $V = \lambda I_D$ and arbitrary $W$.

In the following, for each subset $H \subseteq \mathbb{R}^D$, we denote by $\mathbf{conv}(H)$ the convex hull of $H$, which is the smallest convex set containing $H$ in $\mathbb{R}^D$. For a point $u$, the notation $\mathbf{d}(u, H)$ represents the Euclidean distance from $u$ to $H$, which is defined as

$$\mathbf{d}(u, H) = \inf_{v \in H} \|u - v\|.$$

If $H$ is a closed half-space of $\mathbb{R}^D$ with an outer normal vector $\mathbf{n}$ and $u \notin H$, then

$$\mathbf{d}(u, H) = \mathbf{n}^\top \cdot (u - \mathbf{proj}_H(u)),$$

where $\mathbf{proj}_H(u)$ is the projection of $u$ onto $H$.

We begin with the following lemma, whose proof can be found in [17].

**Lemma B.7.** *Let $H$ be a closed half-space of $\mathbb{R}^D$ with an outer unit normal vector $\mathbf{n}$. Let $u_1, \dots, u_L \colon [0, +\infty) \to \mathbb{R}^D$ be an arbitrary sequence of differentiable functions. For each $t \in [0, +\infty)$, set*

$$\mathbf{Min}(t) = \{1 \leq l \leq L \mid \mathbf{d}(u_l(t), H) = \min_{1 \leq i \leq L} \mathbf{d}(u_i(t), H)\}.$$

*Then we have*

$$\frac{d}{dt} \min_{1 \leq i \leq L} \mathbf{d}(u_i(t), H) = \min_{i \in \mathbf{Min}(t)} \left( \mathbf{n}^\top \cdot \frac{d}{dt}u_i(t) \right).$$

*Proof.* This lemma is already proved in the proof of [17, Proposition 8.2]. $\qquad\square$

The following proposition refines [17, Proposition 8.2], where the condition $V = W = I_D$ was assumed. Here, we extend the proof by relaxing this condition and demonstrating that the result remains valid for matrices of the form $V = \lambda I_D$ with $\lambda > 0$, without imposing any constraints on $W$. Another proof of this proposition can also be found in [21, Proposition 2.1].

**Proposition B.8.** *Assume that $V = \lambda I_D$ for some real number $\lambda > 0$ and $W$ is an arbitrary matrix in $\mathbb{R}^{D \times D}$. Let $(x_1(t), \ldots, x_L(t)) \in C^\infty([0, +\infty)])^L$ be the unique solution of the dynamical system* (9). *Then*

$$e^{-|\lambda|t} x_l(t) \in \mathbf{conv}\left(\{x_{i0}\}_{i=1}^L\right),$$

*for all $l = 1, \ldots, L$ and $t \in [0, +\infty)$.*

*Proof.* Let $z_l(t) = e^{-\lambda t} x_l(t)$. Then we have

$$\frac{d}{dt} z_l(t) = e^{-\lambda t} \left( \frac{d}{dt} x_l(t) - \lambda x_l(t) \right)$$

$$= e^{-\lambda t} \left( \lambda \sum_{i=1}^L \frac{e^{e^{2\lambda t} z_l^\top \cdot W \cdot z_i}}{\sum_{j=1}^L e^{e^{2\lambda t} z_l^\top \cdot W \cdot z_i}} e^{\lambda t} z_i(t) - \lambda e^{\lambda t} z_l(t) \right)$$

$$= \lambda \sum_{i=1}^L \left( \frac{e^{e^{2\lambda t} z_l^\top \cdot W \cdot z_i}}{\sum_{j=1}^L e^{e^{2\lambda t} z_l^\top \cdot W \cdot z_i}} \right) (z_i(t) - z_l(t)).$$

Therefore, the function $(z_1(t), \ldots, z_L(t))$ satisfies the dynamical system:

$$\frac{dz_l(t)}{dt} = \sum_{i=1}^L P_{l,i}(t, z_1(t), \ldots, z_L(t)) \cdot (z_i(t) - z_l(t)), \quad l = 1, \ldots, L, \tag{14}$$

with the initial conditions $z_l(0) = x_{l0}$, where

$$P_{l,i}(t, z_1, \ldots, z_L) = \frac{e^{e^{2\lambda t} z_l^\top \cdot W \cdot z_i}}{\sum_{j=1}^L e^{e^{2\lambda t} z_l^\top \cdot W \cdot z_i}} \cdot \lambda.$$

We claim that, for every closed half-space $H$ of $\mathbb{R}^D$ such that $\mathbf{conv}(\{x_{i0}\}_{i=1}^L) \cap H = \emptyset$, the map $\alpha \colon [0, +\infty) \to \mathbb{R}$ defined by

$$\alpha(t) = \min_{1 \leq i \leq L} \mathbf{d}(z_i(t), H)$$

is non-decreasing. Indeed, using item $(a)$ of Lemma B.7, we have

$$\frac{d}{dt} \alpha(t) = \min_{i \in \mathbf{Min}(t)} \left( \mathbf{n} \cdot \frac{d}{dt} z_i(t) \right)$$

$$= \min_{i \in \mathbf{Min}(t)} \left( \sum_{i=1}^L P_{l,i}(t, z_1(t), \ldots, z_L(t)) \cdot \mathbf{n}^\top \cdot (z_j(t) - z_i(t)) \right).$$

On the right hand side, we have

$$\mathbf{n}^\top \cdot (z_j(t) - z_i(t)) = \mathbf{n}^\top \cdot (z_j(t) - \mathbf{proj}_H(z_j(t))) - \mathbf{n}^\top \cdot (z_i(t) - \mathbf{proj}_H(z_i(t)))$$
$$+ \mathbf{n}^\top \cdot (\mathbf{proj}_H(z_j(t)) - \mathbf{proj}_H(z_j(t)))$$
$$= \mathbf{d}(z_j(t), H) - \mathbf{d}(z_i(t), H)$$

which is nonnegative since $i \in \mathbf{Min}(t)$. Therefore, $\frac{d}{dt} \alpha(t) \geq 0$ and $\alpha$ is non-decreasing. As a consequence, we have

$$\mathbf{d}(z_l(t), H) \geq \min_{1 \leq i \leq L} \mathbf{d}(x_{l0}, H) > 0,$$

for all $l = 1, \ldots, L$ and $t \in [0, +\infty)$. This means that, $z_l(t)$ is outside $H$ as long as $H \cap \mathbf{conv}(\{x_{l0}\}_{i=1}^L) = \emptyset$. Hence,

$$z_l(t) \in \bigcap_{\substack{H \text{ closed half-space} \\ H \cap \mathbf{conv}(\{x_{l0}\}_{i=1}^L) = \emptyset}} H = \bigcap_{\substack{H' \text{ open half-space} \\ H' \supset \mathbf{conv}(\{x_{l0}\}_{i=1}^L)}} H' = \mathbf{conv}\left(\{x_{l0}\}_{i=1}^L\right).$$

The proposition is then proved. $\qquad\square$

In the following, for each nonzero vector $\mathbf{n} \in \mathbb{R}^D$, we denote by $H_{\mathbf{n}}$ the closed half-space of $\mathbb{R}^D$ with normal vector $\mathbf{n}$ such that zero belongs to its boundary $\partial H_{\mathbf{n}}$. In this case, $\partial H_{\mathbf{n}}$ represents the hyperplane containing zero with normal vector $\mathbf{n}$.

**Theorem B.9.** *Assume that the value matrix $V$ has at least one positive eigenvalue. Let $\mathbf{n}$ be an eigenvector of $V$ corresponding to a positive eigenvalue. Then*

$$\min_{1 \leq i \leq L} \left( \mathbf{n}^\top x_{i0} \right) \leq \mathbf{n}^\top e^{-tV^\top} x_l(t) \leq \max_{1 \leq i \leq L} \left( \mathbf{n}^\top x_{i0} \right),$$

*for all $t \in [0, +\infty)$ and $l = 1, \ldots, L$.*

*As a consequence, if the initial points $x_{10}, \ldots, x_{L0}$ are all on one side of the hyperplane $\partial H_{\mathbf{n}}$, and if $V$ has only positive eigenvalues, then*

$$\lim_{t \to +\infty} \|x_l(t)\| = +\infty, \quad \text{for all } l.$$

*Proof.* Let $z_l(t) = e^{-tV^\top} x_l(t)$. Then the function $(z_1(t), \ldots, z_L(t))$ satisfies the dynamical system:

$$\frac{dz_l(t)}{dt} = \sum_{i=1}^{L} P_{l,i}(t, z_1(t), \ldots, z_L(t)) \cdot V^\top \cdot (z_i(t) - z_l(t)), \quad l = 1, \ldots, L, \tag{15}$$

with the initial conditions $z_l(0) = x_{l0}$, where

$$P_{l,i}(t, z_1, \ldots, z_L) = \frac{e^{z_l^\top \cdot e^{tV} \cdot W \cdot e^{tV^\top} \cdot z_i}}{\sum_{j=1}^{L} e^{z_l^\top \cdot e^{tV} \cdot W \cdot e^{tV^\top} \cdot z_j}}.$$

Let $\lambda > 0$ be the eigenvalue associated to $\mathbf{n}$. By multiplying $\mathbf{n}^\top$ into both sides of equation (15), and set $y_l(t) = \mathbf{n}^\top \cdot z_l(t)$, we obtain

$$\frac{dy_l(t)}{dt} = \sum_{i=1}^{L} P_{l,i}(t, z_1(t), \ldots, z_L(t)) \cdot \lambda \cdot (y_i(t) - y_l(t)), \quad l = 1, \ldots, L, \tag{16}$$

with the initial conditions $y_l(0) = \mathbf{n}^\top \cdot x_{l0} \in \mathbb{R}$. By using the same argument in Proposition B.8, with $z_l$ is replaced by $y_l$ here, we see that

$$y_l(t) \in \mathbf{conv}\left( \{y_{i0}\}_{i=1}^{L} \right),$$

and hence,

$$\min_{1 \leq i \leq L} y_{i0} \leq y_l(t) \leq \max_{1 \leq i \leq L} y_{i0},$$

for all $t \in [0, +\infty)$ and $l = 1, \ldots, L$. Therefore,

$$\min_{1 \leq i \leq L} \left( \mathbf{n}^\top \cdot x_{i0} \right) \leq \mathbf{n}^\top \cdot e^{-tV^\top} \cdot x_l(t) \leq \max_{1 \leq i \leq L} \left( \mathbf{n}^\top \cdot x_{i0} \right),$$

as claimed.

In case the initial points $x_{10}, \ldots, x_{L0}$ are all one side of the hyperplane $\partial H_{\mathbf{n}}$, then either $\min_{1 \leq i \leq L} \left( \mathbf{n}^\top \cdot x_{i0} \right) > 0$ or $\max_{1 \leq i \leq L} \left( \mathbf{n}^\top \cdot x_{i0} \right) < 0$. In both case, there is $\epsilon > 0$ such that

$$\left| \mathbf{n} \cdot e^{-tV^\top} \cdot x_l(t) \right| > \epsilon$$

for all $t \in [0, +\infty)$ and $l$. Hence, when $V$ has only positive eigenvalues, we must have

$$\lim_{t \to +\infty} \|x_l(t)\| = +\infty, \quad \text{for all } l.$$

The theorem is then proved. □

## C   Effects of Absolute and Rotary Positional Encodings

In this section, we analyze the influence of absolute and rotary positional encodings on the dynamical behavior of tokens within self-attention mechanisms.

## C.1 Absolute Positional Encoding

Recall that the dynamical system governing the continuous-time limit of the self-attention with absolute positional encoding is given by

$$\frac{dx_l(t)}{dt} = V^\top \cdot \sum_{i=1}^{L} \left( \frac{e^{(x_l(t)+p_l)^\top \cdot W \cdot (x_i(t)+p_i)}}{\sum_{j=1}^{L} e^{(x_l(t)+p_l)^\top \cdot W \cdot (x_j(t)+p_j)}} \right) \cdot (x_i(t) + p_i), \qquad (17)$$

for $l = 1, \ldots, L$, with the initial conditions

$$(x_1(0), \ldots, x_L(0)) = (x_{10}, \ldots, x_{L0}) \in (\mathbb{R}^D)^L.$$

Here $p_i = [p_{i,1}, \ldots, p_{i,D}]^\top \in \mathbb{R}^D$. A common choice is to either learn the positional embeddings $p_i$ jointly with the model parameters, or to fix them using a sinusoidal scheme: with

$$p_{i,j} = \begin{cases} \sin(i \cdot 10000^{-\frac{j}{D}}), & \text{if } j \text{ is even,} \\ \cos(i \cdot 10000^{-\frac{j-1}{D}}), & \text{if } j \text{ is odd.} \end{cases}$$

This differential system can be easily transformed into equation (2) by the transition $x_l \mapsto x_l + p_l$. Therefore, the dynamical properties of the self-attention with and without absolute are the similar as we will see in the following corollaries.

**Corollary C.1.** *Let $(x_1(t), \ldots, x_L(t)) \in C^\infty([0, +\infty))$ be a solution of the dynamical system (17). If $A \prec 0$ and $W \succ 0$, then $\lim_{t \to +\infty} x_l(t) = -p_l$ for all $l = 1, \ldots, L$.*

**Corollary C.2.** *Let $X(t) = (x_1(t), \ldots, x_L(t)) \in C^\infty([0, +\infty))^L$ be a solution of the differential system (17). Assume that the value matrix $V$ has at least one positive eigenvalue. Let $\mathbf{n}$ be an eigenvector of $V$ corresponding to a positive eigenvalue. Then*

$$\min_{1 \le i \le L} \left( \mathbf{n}^\top (x_{i0} + p_i) - e^{-tV^\top} p_l \right) \le \mathbf{n}^\top e^{-tV^\top} x_l(t) \le \max_{1 \le i \le L} \left( \mathbf{n}^\top (x_{i0} + p_i) \right) - e^{-tV^\top} p_l,$$

*for all $t \in [0, +\infty)$ and $l = 1, \ldots, L$.*

*As a consequence, if the points $x_{10} + p_1, \ldots, x_{L0} + p_L$ are all on one side of the hyperplane $\partial H_\mathbf{n} + e^{-tV^\top} p_l$, and if $V$ has only positive eigenvalues, then*

$$\lim_{t \to +\infty} \|x_l(t)\| = +\infty, \quad \text{for all } l.$$

## C.2 Rotary Positional Encoding

In this section, we assume that the token dimension $D$ is an even number. In contrast to absolute positional encoding, the continuous-time dynamics of the self-attention with rotary positional encoding can be described via the differential system [46]

$$\frac{dx_l(t)}{dt} = V^\top \cdot \sum_{i=1}^{L} \left( \frac{e^{x_l(t)^\top \cdot W_{li} \cdot x_i(t)}}{\sum_{j=1}^{L} e^{x_l(t)^\top \cdot W_{lj} \cdot x_j(t)}} \right) \cdot x_i(t), \qquad (18)$$

for $l = 1, \ldots, L$, where

$$W_{li} = \frac{1}{\sqrt{D_k}} \left( Q \cdot K^\top + \overline{Q} \cdot R^D_{\theta, i-l} \cdot \overline{K}^\top \right), \qquad (19)$$

with $Q, K, \overline{Q}, \overline{K} \in \mathbb{R}^{D \times D}$ are two additional learnable matrices and

$$R^D_{\Theta, m} = \begin{pmatrix} \cos m\theta_1 & -\sin m\theta_1 & 0 & 0 & \cdots & 0 & 0 \\ \sin m\theta_1 & \cos m\theta_1 & 0 & 0 & \cdots & 0 & 0 \\ 0 & 0 & \cos m\theta_2 & -\sin m\theta_2 & \cdots & 0 & 0 \\ 0 & 0 & \sin m\theta_2 & \cos m\theta_2 & \cdots & 0 & 0 \\ \vdots & \vdots & \vdots & \vdots & \ddots & \vdots & \vdots \\ 0 & 0 & 0 & 0 & \cdots & \cos m\theta_{D/2} & -\sin m\theta_{D/2} \\ 0 & 0 & 0 & 0 & \cdots & \sin m\theta_{D/2} & \cos m\theta_{D/2} \end{pmatrix}$$

and $\Theta = \{\theta_i = 10000^{-2(i-1)/D}, \ i \in [1, 2, \ldots, D/2]\}$.

Rotary positional encoding is an essential component of latent attention, which lies at the core of DeepSeek [31]. Unlike absolute positional encoding, the differential system governing rotary positional encoding exhibits markedly different behavior. In cases where token trajectories diverge to infinity under absolute positional encoding, we observe a similar divergence scenario in the presence of rotary positional encoding, as demonstrated below:

**Corollary C.3.** *Let $X(t) = (x_1(t), \ldots, x_L(t)) \in C^\infty([0, +\infty))^L$ be a solution of the differential system* (18). *Assume that the value matrix $V$ has at least one positive eigenvalue. Let $\mathbf{n}$ be an eigenvector of $V$ corresponding to a positive eigenvalue. Then*

$$\min_{1 \le i \le L} \left( \mathbf{n}^\top x_{i0} \right) \le \mathbf{n}^\top e^{-tV^\top} x_l(t) \le \max_{1 \le i \le L} \left( \mathbf{n}^\top x_{i0} \right),$$

*for all $t \in [0, +\infty)$ and $l = 1, \ldots, L$.*

*As a consequence, if the points $x_{10}, \ldots, x_{L0}$ are all on one side of the hyperplane $\partial H_{\mathbf{n}}$, and if $V$ has only positive eigenvalues, then*

$$\lim_{t \to +\infty} \|x_l(t)\| = +\infty, \quad \text{for all } l.$$

*Proof.* Apply the same argument as in the proof of Theorem B.9. □

**Remark C.4.** In contrast to absolute positional encoding, rotary positional encoding induces notably different dynamical behavior by promoting token divergence (even for the cases when tokens converge to a finite point in the absolute positional encoding). Specifically, the presence of the additional term $\overline{Q} \cdot R^D_{\theta, i-l} \cdot \overline{K}^\top$ in the query-key interaction matrix $W_{li}$, as defined in equation (19), can hinder the system from transitioning into a convergence regime. Consequently, self-attention equipped with rotary positional encoding tends to exhibit divergence behavior more frequently than those using absolute encoding or no positional encoding at all.

## D  Additional Experimental Details

### D.1  Parameter settings used in the simulations

#### D.1.1  Figure 1: Distances between tokens over time

On the left side of Figure 1, we choose

- $A = \begin{pmatrix} 1.72628 & -3.79592 \\ -0.914069 & 3.49779 \end{pmatrix}$ whose symmetric component $A_{\text{sym}}$ has positive eigenvalues 5.12809 and 0.0959758;
- $W = \begin{pmatrix} 0.534636 & -0.798866 \\ -1.17152 & -1.92153 \end{pmatrix}$; and
- the initial values $x_{10} = (-1.17525, 1.99834)$, $x_{20} = (-0.0231564, 0.591678)$, $x_{30} = (-0.94811, -1.37996)$, $x_4 = (1.00246, -1.69335)$.

On the right side of Figure 1, we choose

- $A = \begin{pmatrix} -1.43778 & -1.10989 \\ 0.563455 & -0.401696 \end{pmatrix}$ whose symmetric component $A_{\text{sym}}$ has negative eigenvalues $-1.50541$ and $-0.334061$;
- $W = \begin{pmatrix} 0.433083 & -0.0371911 \\ -0.715343 & -1.53568 \end{pmatrix}$; and
- the initial values $x_{10} = (0.123688, 0.20691)$, $(x_{20} = (0.53086, 1.47281)$, $x_{30} = (-0.78388, -1.24115)$, $x_{40} = (1.63476, 0.321809)$.

#### D.1.2  Figure 2: Token trajectories under convergence and divergence scenarios

In Figure 2a, we choose the following parameters:

- $A = \begin{pmatrix} -2.94058 & -2.12076 \\ -5.14498 & -4.58104 \end{pmatrix}$, thus the symmetric component $A_{\text{sym}}$ has negative eigenvalues $-7.48513$ and $-0.0364942$;
- $W = \begin{pmatrix} 0.902496 & -2.37879 \\ 4.36478 & 3.84768 \end{pmatrix}$, thus the symmetric component $W_{\text{sym}}$ has positive eigenvalues 4.15119 and 0.598979.

Figure 2b illustrates the divergence scenario with $A = 2W$ (thus $V = 2I$) and

$$W = \begin{pmatrix} -0.404078 & 0.982735 \\ -0.567909 & 0.600242 \end{pmatrix}.$$

### D.1.3 Figure 5: Token trajectories shift to divergence regime in RoPE

On the left of Figure 5, we choose

- $Q = \begin{pmatrix} 0.07331137 & 0.17647239 \\ -0.32738218 & -0.43457359 \end{pmatrix}$

- $K = \begin{pmatrix} -2.54009796 & 1.82991692 \\ -0.95688637 & 0.60349328 \end{pmatrix}$, thus $W_{\text{sym}}$ have positive eigenvalues 0.03766541 and 0.15005164

- $V = -1.5I$

  On the right of Figure 5, we choose the additional parameters

  - $\overline{Q} = \begin{pmatrix} -3.01517413 & 2.4430872 \\ 2.11630464 & 1.40111342 \end{pmatrix}$

  - $\overline{K} = \begin{pmatrix} 5.03454859 & -3.12492845 \\ 4.58643881 & -2.00780098 \end{pmatrix}$,

### D.1.4 Figure 9: Token trajectories in convergence and divergence regime

On Figure 9a (convergence case), we choose the following parameters:

- $Q = \begin{pmatrix} -1.18765511 & 0.8975229 \\ -0.7793589 & 0.79105257 \end{pmatrix}$

- $K = \begin{pmatrix} -1.97520362 & -1.98198651 \\ 2.24167927 & 2.93460903 \end{pmatrix}$

- $\overline{Q} = \begin{pmatrix} 1.04027991 & -0.12991073] \\ -1.32542484 & 1.08074871 \end{pmatrix}$

- $\overline{K} = \begin{pmatrix} -1.00477795 & -0.48804888 \\ -0.42151108 & 0.02556926 \end{pmatrix}$

- $V = -I$

On Figure 9b (divergence case), we choose the following parameters:

- $Q = \begin{pmatrix} 2.068739 & -1.83750201 \\ -0.75622145 & 0.4784381 \end{pmatrix}$

- $K = \begin{pmatrix} -1.12583337 & -1.40120114 \\ -2.79629618 & -3.24668939 \end{pmatrix}$

- $\overline{Q} = \begin{pmatrix} -0.67880222 & 1.21234986 \\ -0.67132474 & 0.90406252 \end{pmatrix}$

- $K = \begin{pmatrix} 0.57406142 & 2.88899216 \\ -1.10421806 & 0.75603913 \end{pmatrix}$

- $V = 1.5I$

### D.2 Implementation of Convergence, Divergence, and Intermediate scenarios.

We propose a strategy to guarantee that the model falls into the three scenarios described in Section 5.1 and Section 3.4. In particular, we introduce a reparameterization of the matrices $Q$, $K$, and $V$ that enforces the positive or negative definiteness of the matrices

$$W_{\text{sym}} = \frac{1}{2}\left(W + W^\top\right), \quad A_{\text{sym}} = \frac{1}{2}\left(A + A^\top\right), \tag{20}$$

where $W = QK^\top$ and $A = W(V^\top)^{-1}$. We first describe the approach to guarantee the positive definiteness of $A_{\text{sym}}$ and $W_{\text{sym}}$ and subsequently extend the framework to accommodate negative definiteness and intermediate cases.

**Ensuring the Positive Definiteness of $A_{\mathbf{sym}}$.** To enforce the positive definiteness of $A_{\text{sym}}$, we leverage the $LDL^T$ decomposition [18]. Specifically, we parametrize:

$$A_{\text{sym}} = L_a D_a L_a^\top, \tag{21}$$

where $L_a$ is a lower triangular matrix with ones on the diagonal, and $D_a$ is a diagonal matrix with strictly positive elements. The positivity of $D_a$ is ensured by applying the Softplus function:

$$D_a = \text{diag}(\text{Softplus}(d_a)). \tag{22}$$

**Parametrization of $A$ and $W$.** Given $A_{\text{sym}}$, from Eq. 20 we obtain

$$A = A_{\text{sym}} + X_a, \tag{23}$$

where $X_a$ is an antisymmetric matrix satisfying $X_a = -X_a^\top$. A natural parametrization for such a matrix is:

$$X_a = T_a - T_a^\top, \tag{24}$$

Consequently, the final parametrization of $A$ ensuring positive definiteness of $A_{\text{sym}}$ is

$$A = L_a \text{diag}(\text{Softplus}(d_a))L_a^\top + T_a - T_a^\top. \tag{25}$$

A similar parametrization applies to $W$ to enforce the positive definiteness of $W_{\text{sym}}$:

$$W = L_w \text{diag}(\text{Softplus}(d_w))L_w^\top + T_w - T_w^\top. \tag{26}$$

The above formulations ensures that both $W_{\text{sym}}$ and $A_{\text{sym}}$ maintain the desired definiteness properties while allowing for a flexible parametrization of $W$ and $A$.

**Parametrization of $Q$, $K$, and $V$.** In terms of $Q$, $K$, and $V$, the aforementioned formulations equivalent to:

$$\begin{cases} Q \cdot K^\top & = L_w \text{diag}(\text{Softplus}(d_w))L_w^\top + T_w - T_w^\top, \\ Q \cdot K^\top \cdot \left(V^\top\right)^{-1} & = L_a \text{diag}(\text{Softplus}(d_a))L_a^\top + T_a - T_a^\top. \end{cases} \tag{27}$$

We designate $Q$, $L_w$, $d_w$, $L_a$, $d_a$, $T_w$, and $T_a$ as learnable parameters. The key and value projection matrices for self-attention are then computed as:

$$\begin{cases} K & = \left[Q^{-1} \cdot L_w \text{diag}(\text{Softplus}(d_w))L_w^\top + T_w - T_w^\top\right]^\top, \\ V & = \left[\left(L_a \text{diag}(\text{Softplus}(d_a))L_a^\top + T_a - T_a^\top\right)^{-1} \cdot \left(L_w \text{diag}(\text{Softplus}(d_w))L_w^\top + T_w - T_w^\top\right)\right]^\top. \end{cases} \tag{28}$$

This parametrization guarantees that $A_{\text{sym}}$ and $W_{\text{sym}}$ are positive definite.

**Adjustments for Negative Definite and Intermediate Cases.** The definiteness of $A_{\text{sym}}$ and $W_{\text{sym}}$ is determined by the sign of the elements in $D_a$ and $D_w$, respectively. In the positive definite case, these elements are constrained to be positive using the Softplus function. To adapt to the negative definite and intermediate cases, we control the sign of the elements in $D_a$ and $D_w$ by multiplying the output of Softplus with a vector containing only $-1$ for the negative definite case or a vector containing an equal number of $1$ and $-1$ for the intermediate case.

This approach ensures that our parametrization is flexible enough to accommodate different definiteness requirements while maintaining learnability and numerical stability.

### D.3 Details on Language Modeling experiments

### D.3.1 Dataset

**WikiText-103.** [34] The WikiText-103 dataset comprises approximately 268,000 unique words. Its training set includes around 28,000 articles, totaling 103 million tokens. On average, this corresponds to text blocks of about 3,600 words per article. The validation and test sets each consist of 60 articles, containing 218,000 and 246,000 tokens, respectively.

**EnWik8.** [20] The Enwik8 dataset is a byte-level corpus comprising 100 million bytes extracted from Wikipedia. In addition to standard English text, it includes markup, special characters, and content in multiple languages. The standard split provides 90 million bytes for training and 5 million for testing.

### D.3.2 Wikitext103 Model and Training Configurations

**Model.** We utilize the Transformer-XL [9] (https://github.com/kimiyoung/transformer-xl) architecture for word-level language modeling on the WikiText-103 dataset. The model comprises 16 layers, each with a hidden size of 410. Multi-head attention is implemented with 10 heads, each having a dimensionality of 41. The position-wise feedforward networks have an inner dimension of 2100. Regularization is applied via a dropout rate of 0.05 on residual connections. For Rotary positional embedding, the rotational dimension used is 16.

**Training Configurations.** Training is conducted using the Adam optimizer with a learning rate of 0.00025. A linear warmup is applied for the first 1,000 steps, followed by a cosine annealing schedule over a total of 200,000 training steps. The model is trained with a target sequence length of 150 tokens and no memory length, effectively disabling the segment-level recurrence mechanism. Evaluation is performed with a slightly longer target length of 156 tokens. Training utilizes 2 NVIDIA A100 SXM4 80GB GPUs with a total batch size of 60.

### D.3.3 EnWik8 Model and Training Configurations

**Model.** We trained an autoregressive Transformer model on the Enwik8 dataset using the x-transformers (https://github.com/lucidrains/x-transformers) library. The model follows a GPT-style architecture, comprises a 6-layer Transformer decoder. Each layer uses 8 attention heads and a model dimension of 512. For tokenization, byte-level encoding was used, resulting in a vocabulary size of 256 unique tokens. Both training and generation sequences were fixed at 1024 tokens.

**Training Configurations.** Data was sampled into overlapping sequences of length 1025 (1024 input tokens plus 1 target token). We used the Adam optimizer with a learning rate of $\times 10^{-4}$, and applied gradient clipping with a maximum norm of 0.5. Gradients were accumulated over 4 steps to simulate larger batch sizes. A batch size of 4 was used, with gradient accumulation yielding an effective batch size of 16. The model was trained for 100,000 iterations. Validation was performed every 100 steps, and text samples were generated every 500 steps with a generation length of 1024 tokens. Training was performed on a NVIDIA A100 SXM4 80GB GPU.

## D.4 Details on ImageNet-1K object recognition task

### D.4.1 Dataset

**ImageNet-1K.** [11] This dataset spans 1000 object classes and contains 1,281,167 training images, 50,000 validation images. The model learns to predict the class of the input image among 1000 categories. We report the top-1 and top-5 accuracy on all experiments.

### D.4.2 Model and Training Configurations

**Models.** We adopted the DeiT [50] (https://github.com/facebookresearch/deit) architecture, a lightweight Vision Transformer (ViT) variant designed for efficient image classification. The model used is Tiny version, which divides each 224×224 input image into non-overlapping 16×16 patches, resulting in a sequence of 196 tokens. Each patch is linearly projected into a 192-dimensional embedding space. The Transformer encoder consists of 12 layers, each employing multi-head self-attention with 3 heads, and an MLP block with a hidden dimension four times the embedding size (i.e., 768). Biases are included in the query, key, and value projections, and layer normalization is applied with an epsilon value of 1e-6. The model includes a learnable [CLS] token and absolute positional embeddings. No distillation token or teacher model was used, and no architectural modifications (e.g., convolutional stems or hybrids) were introduced.

**Training Configurations.** The model was trained on the full ImageNet-1k training set for 300 epochs using the AdamW optimizer with a base learning rate of $5 \times 10^{-4}$, and weight decay of 0.05. The learning rate followed a cosine decay schedule, with a linear warmup phase over the first 5 epochs, and a minimum learning rate of $10^{-5}$. A stochastic depth rate (drop path) of 0.1 were used for regularization. Data augmentation included RandAugment, Mixup with $\alpha = 0.8$, CutMix with $\alpha = 1.0$, label smoothing with $\epsilon = 0.1$, and random erasing with a probability of 0.25. The model was trained with a batch size of 64 across 4 NVIDIA A100 SXM4 80GB GPUs using mixed precision. Exponential Moving Average (EMA) of model weights was maintained with a decay factor of 0.99996.

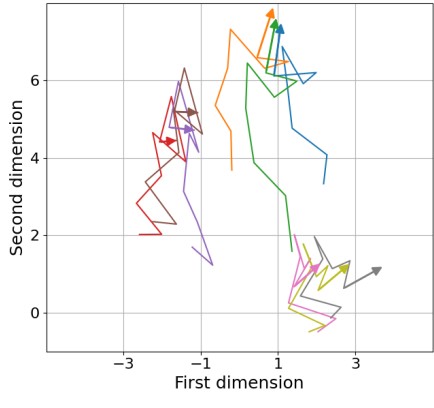

(a) With LayerNorm, without feedforward.

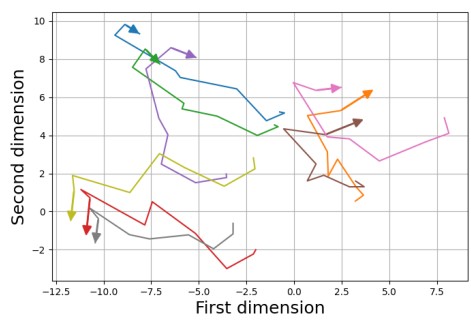

(b) With both LayerNorm and feedforward.

Figure 8: Tokens' trajectory in the later layers of a pre-trained GPT-2 model forms clusters with similar shapes.

### D.5 Additional visualizations

#### D.5.1 Tokens' trajectory in pre-trained Transformers

We provide additional visualization on pre-trained GPT-2 model to observe the trajectory of tokens. As illustrated in Figure 8, tokens trajectories in later layers still forms clusters with similar shapes.

#### D.5.2 Token trajectories with RoPE

We provide additional simulations of token trajectories in the system of Equation 8 for the convergence (Figure 9a) and divergence case (Figure 9b).

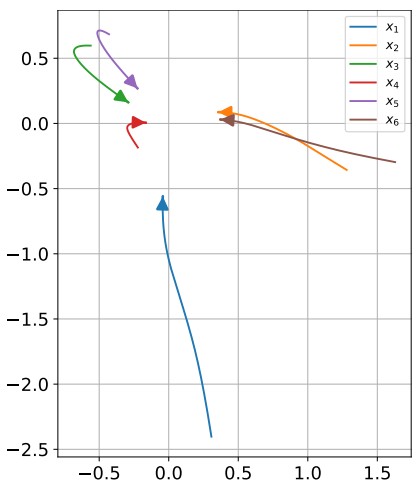

(a) Convergence case of Self-Attention + RoPE. Tokens tend to converge to zero when $t \to \infty$.

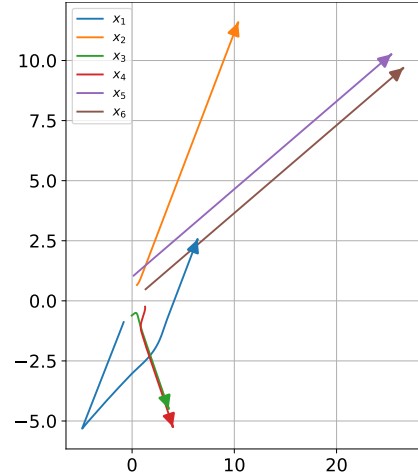

(b) Divergence case of Self-Attention + RoPE. Tokens tend to diverge to $\infty$ when $t \to \infty$.

Figure 9: Self-Attention with RoPE under (a) convergence and (b) divergence settings.

### D.6 Additional Results

#### D.6.1 Full Evaluation Results

In this section, we demonstrate the means and standard deviations of the experiments we executed. Table 4, 5, 6 show the means and standard deviations of our experiments in Section 5.2, Section 5.1.

#### D.6.2 Condition verification on pretrained Transformers

We performed an empirical analysis of real-world pretrained Transformer models to verify the conditions in our theory. Specifically, for each model, we measured the mean percentage across

Table 4: Bits Per Characters (BPC) and Perplexity (PPL) of Transformers with Rotary Positional Encoding across different scenarios on EnWik8 and WikiText-103 language modeling.

| Scenario | EnWik8 Pretrain Test BPC ($\downarrow$) | WikiText-103 Pretrain | |
| --- | --- | --- | --- |
| | | Valid PPL ($\downarrow$) | Test PPL ($\downarrow$) |
| *Transformer + RoPE* | $1.295 \pm 0.003$ | $31.37 \pm 0.14$ | $32.35 \pm 0.18$ |
| Transformer + RoPE + $\lambda I_D$ | $1.288 \pm 0.002$ | $31.10 \pm 0.16$ | $32.09 \pm 0.17$ |
| Transformer + RoPE + $\lambda A$ | $\mathbf{1.281 \pm 0.002}$ | $\mathbf{31.06 \pm 0.11}$ | $\mathbf{32.04 \pm 0.14}$ |

Table 5: Bits Per Characters (BPC) and Perplexity (PPL) of Transformers with sinusoidal positional encoding across scenarios on EnWik8 and WikiText-103 language modeling.

| Scenario | EnWik8 Pretrain Test BPC ($\downarrow$) | WikiText-103 Pretrain | |
| --- | --- | --- | --- |
| | | Valid PPL ($\downarrow$) | Test PPL ($\downarrow$) |
| *Baseline* | $1.331 \pm 0.002$ | $31.63 \pm 0.12$ | $32.37 \pm 0.10$ |
| Convergence | $1.350 \pm 0.003$ | $32.24 \pm 0.15$ | $33.05 \pm 0.13$ |
| Intermediate | $1.345 \pm 0.002$ | $31.91 \pm 0.11$ | $32.78 \pm 0.12$ |
| Divergence | $\mathbf{1.324 \pm 0.002}$ | $\mathbf{31.12 \pm 0.11}$ | $\mathbf{32.07 \pm 0.13}$ |

Table 6: Top 1 and Top 5 Validation Accuracy of DeiT (learnable positional encoding) across different scenarios on ImageNet1K.

| | Top-1 Acc ($\uparrow$) | Top-5 Acc ($\uparrow$) |
| --- | --- | --- |
| *DeiT baseline* | $71.79 \pm 0.02$ | $90.99 \pm 0.02$ |
| + convergence | $71.37 \pm 0.02$ | $90.65 \pm 0.01$ |
| + intermediate | $71.62 \pm 0.03$ | $90.94 \pm 0.02$ |
| + divergence | $71.93 \pm 0.02$ | $91.03 \pm 0.03$ |

all layers of (i) "near-zero" eigenvalues of the value matrix $V$ (using threshold $\epsilon = 10^{-3}$), and (ii) positive eigenvalues of the symmetrized matrices $W_{\mathrm{sym}}$ and $A_{\mathrm{sym}}$.

Table 1 reports the mean $\pm$ standard deviation of these percentages for GPT-2 XL, DistilGPT2, and LLaMA-2 13B. We observe that: (i) In all cases, $V$ exhibited 0% near-zero eigenvalues, indicating that FGPT$V$ is invertible in practice. (ii) The proportion of positive eigenvalues in both $W_{\mathrm{sym}}$ and $A_{\mathrm{sym}}$ is approximately 50%, which matches the divergence regime predicted by Remark 3.5 if omitting FFNs and LayerNorms. The divergence behaviour of a pretrained model is also illustrated in 'baseline' case of Figure 6 in Section 5.1.

These findings align with theoretical expectations: (i) the set of singular matrices has measure zero in continuous parameter spaces, and (ii) unconstrained weight matrices, whether randomly initialized or learned, tend to have eigenvalue distributions symmetric about zero. We also repeated this analysis on a randomly initialized model (GPT-2 XL reinit) and obtained similar results (approximately 50% positive eigenvalues in $W_{\mathrm{sym}}$, $A_{\mathrm{sym}}$, and no near-zero eigenvalues in $V$).

### D.6.3 Layerwise Token Norms and Pairwise Distances in GPT-2

We measured the distance between tokens and token norms across layers using test sequences consist of 100 tokens from WikiText-103. Two variants considered: simplified (GPT-2 with no FFNs or LayerNorms) and full GPT-2 models.

In the simplified model (Figures 10a and 10b), both the mean distance between tokens and the mean token norm grow exponentially across layers, consistent with the behavior of the divergence scenario.

In contrast, for the full GPT-2 model (which includes FFNs and LayerNorms), we observe that both token norms and distances increase in the early layers but plateau or decline in the later layers (see Figures 10c and 10d). This behavior deviates from our theoretical predictions, indicating that our result does not fully hold in the presence of LayerNorms and FFNs. A comprehensive theoretical and empirical analysis of these effects is a promising direction for future work.

Table 7: Mean $\pm$ std. dev. of eigenvalue statistics (in %).

| Model | % positive eigval of $W_{\mathrm{sym}}$ | % positive eigval of $A_{\mathrm{sym}}$ | % near-zero-eig of $V$ |
|---|---|---|---|
| GPT2-xl | $50.20 \pm 3.76$ | $49.99 \pm 0.06$ | $0.00 \pm 0.00$ |
| DistilGPT2 | $46.01 \pm 4.36$ | $49.96 \pm 0.16$ | $0.00 \pm 0.00$ |
| Llama2 13B | $52.35 \pm 2.84$ | $50.00 \pm 0.02$ | $0.00 \pm 0.00$ |
| GPT2-xl reinit | $50.20 \pm 2.85$ | $49.99 \pm 0.06$ | $0.00 \pm 0.00$ |

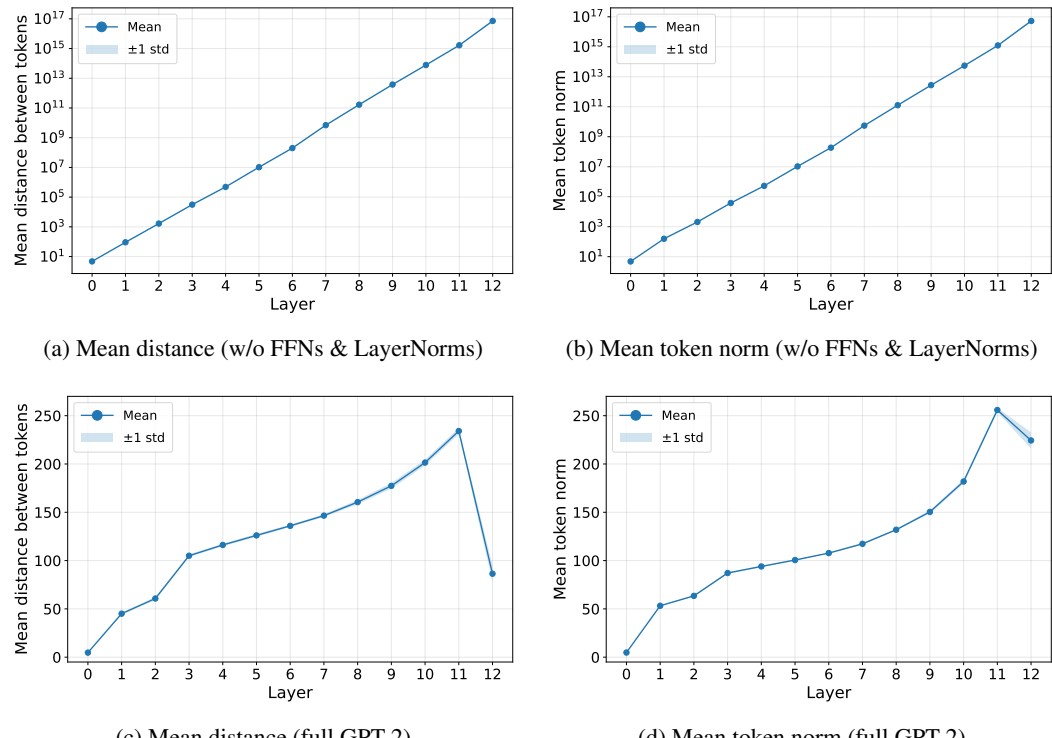

(a) Mean distance (w/o FFNs & LayerNorms)   (b) Mean token norm (w/o FFNs & LayerNorms)

(c) Mean distance (full GPT-2)   (d) Mean token norm (full GPT-2)

Figure 10: Layerwise token of a 100-token sequence. Top: GPT-2 without FFNs/LayerNorms; Bottom: full GPT-2. Left: mean distance between tokens; Right: mean token norm.

Additionally, we present extended experimental results using a broader range of sequence lengths and a larger set of test sequences to provide a more comprehensive analysis of token behavior. In Figures 11b and 11a, we report the mean token norm and the mean distance between tokens, respectively, for sequence lengths of 16, 64, 100, 128, and 256 tokens. For each length, the reported values are averaged over 100 randomly selected sequences from the test set of WikiText-103.

# E   Broader Impact

This work advances the theoretical understanding of token dynamics in Transformer models, providing insights that could enhance model stability and performance across various applications. While primarily theoretical, these findings may inform the development of more robust AI systems. However, as with any advancement in AI, there is a potential for misuse in applications. We encourage the community to consider these implications and promote responsible use of such technologies.

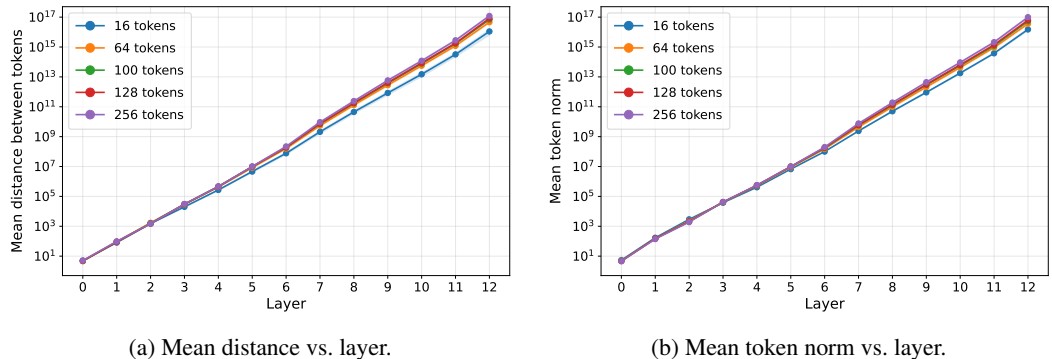

(a) Mean distance vs. layer.    (b) Mean token norm vs. layer.

Figure 11: Sequence-length sensitivity in GPT-2 without FFNs/LayerNorms (100 sequences per length).

