# OpenReview forum: "Dynamical Properties of Tokens in Self-Attention and Effects of Positional Encoding"
_NeurIPS.cc/2025/Conference — NeurIPS 2025 poster_

### Official Review · Reviewer_m1Uq · 2025-06-30

**Clarity:** 3
**Significance:** 2
**Originality:** 2
**Rating:** 4
**Confidence:** 4

**Summary:**

The paper investigates the dynamical behavior of tokens in Transformer models using a Neural-ODE formulation of attention. It examines whether, under certain conditions, tokens tend to converge or diverge as they progress through layers, depending on model parameters. The authors also analyze the role of different positional encodings, specifically absolute and rotary, in shaping these dynamics, and try to support their claims through empirical evaluation.

**Questions:**

**Questions:**

1. I have several questions and suggestions regarding the experimental validation of the paper’s claims:

    a) Is it possible to empirically validate whether tokens in a pretrained model converge or diverge as they propagate through layers? For instance, could this be tested by analyzing distances between token embeddings across layers in a realistic model? Appendix D.5.1 touches on this, but I was hoping for a more thorough empirical investigation using standard pretrained architectures.

    b) Can one analyze the structure of the attention matrices \$QK^\top\$ in real models to determine whether they are PSD, NSD, or neither? In decoder-only models using causal self-attention, these matrices are typically not symmetric, even when assuming \$Q = K\$, so they are unlikely to satisfy PSD or NSD conditions. How would this affect the applicability of the theoretical analysis?

    c) Is it possible to evaluate the core Neural ODE assumption that the model can be approximated by an Euler discretization due to small weight changes between layers? In practice, the \$Q\$, \$K\$, and \$V\$ matrices vary significantly across layers in standard Transformers, which may invalidate this assumption.

    d) Could the experiments in Section 3.4 be rerun without explicitly imposing the constraints used in the theoretical setup? This would help evaluate whether the convergence or divergence behavior still appears under more realistic conditions.

    e) Is it possible to study how convergence or divergence behavior is affected by other architectural components, such as MLP blocks or layer normalization?

    f) Finally, could the analysis be extended to investigate the effect of varying the input sequence length, at least empirically?

2. For what set of models is this analysis even useful? For causal self attention I think a lot of the analysis would probably not hold. Could you please clarify what would change ?

3. In the caption of Figure-4 what does it mean by similar shapes?

4. I think I probably missed this in the paper, but how do you obtain figure 4 and figure 5? Is it via the PCA of the tokens or is there something else ?

5. For Table - 1 and Table -2 is it possible to add any standard deviation

6. In the solution of Equation (11) (see lines 575–578), could the authors clarify the initial conditions used for the token representations $x_\ell(0)$? This question arises because if the quantity $e^{c||x_\ell(0)||^2} - L$ is negative, then the expression may decrease as $t$ increases and eventually become undefined. In such a case, the conclusion that $\lim{t \to +\infty} ||x_\ell(t)||= +\infty$ would no longer hold. This also suggests that the behavior of the system would depend on $t$, meaning that the model's depth (or number of layers) must lie within a bounded domain to ensure validity. If this interpretation is correct, some of the paper's conclusions may need to be reconsidered under such conditions. Could the authors elaborate on what initial values are assumed in these equations, and whether such conditions are expected to occur in practice? Furthermore, for very long input sequences, it appears that $L$ increases with sequence length. In this case, under what conditions on $A$ and $x\ell(0)$ does the term $e^{c||x_\ell(0)||_A^2} - L$ remain positive to ensure the validity of the derivation? That is, how must the matrix $A$ and the initialization be chosen to guarantee that the analysis holds for long sequences?

7. I would like to request clarification regarding the setup used to produce Figure 6. Specifically, is the figure generated by taking a pretrained model and subsequently removing the layer normalization and MLP components, or is it based on a model trained from scratch without using layer norms and MLPs? Additionally, can experiments be conducted where the constraints described in lines 263–272 are removed, in order to observe what happens when these artificial constraints are relaxed?

8. Could the authors elaborate on the intuition behind why divergence leads to improved performance, as suggested in lines 275–279? Is the idea that token representations become sufficiently separated in space, making them easier to distinguish and thus improving model expressivity? This seems somewhat counterintuitive from a language modeling perspective, where one might expect that semantically similar tokens should cluster together, while only dissimilar tokens diverge. Otherwise, it could hinder the model’s ability to learn meaningful attention maps, especially if all tokens diverge indiscriminately. Could the authors provide a more detailed intuition for why divergence is beneficial in this context, and how this compares to what is observed in standard pretrained models?

9. What is the practical benefit of Theorems and Propositions 3.1–3.6? All of these results appear to focus on asymptotic behavior with respect to model depth. However, in practice, the depth is always finite for example, DeepSeek-V3 [A] has approximately 61 layers. Given this, how should we interpret the relevance of these asymptotic results? Do they meaningfully inform us about the behavior of realistic models? More specifically, is there a way to understand how the described properties evolve layer by layer, rather than only in the limit as depth approaches infinity?

**Minor Typos:**

1. Line 179: “We observe from randomly ~~chosen of~~ model parameters”, "chosen" -> "initialized"?


**Note:** I find the analysis in this work to be interesting. However, my main concern is regarding the applicability of the results. Much of the analysis appears to depend on very simplified settings, and the assumptions made are quite strong and seem to diverge significantly from practical scenarios. Based on these concerns, I am currently assigning a lower score. That said, I am open to revisiting my evaluation during the rebuttal phase, depending on the authors' clarifications and the views expressed by other reviewers.

---

**References:**

[A] Liu, Aixin, et al. "Deepseek-v3 technical report." arXiv preprint arXiv:2412.19437 (2024).

**Ethical Concerns:**

["NO or VERY MINOR ethics concerns only"]

**Final Justification:**

I thank the authors for their comments and efforts throughout the rebuttal process. I was on the fence about whether to increase the score for the paper. After weighing the pros and cons, I have decided to raise the score from **2** to **4**.

**Pros:**
1. The authors made substantial efforts during the rebuttal, including running additional experiments and clarifying important points.
2. One of the experiments (the GPT-2 experiments) appears to be consistent with the theoretical findings.

**Cons:**
1. During the discussion phase, we identified a flaw in one of the proofs. While this is not unusual (the rebuttal process often serves to uncover such issues), it will require rewriting portions of the paper. This makes it challenging to judge the quality of the future, revised version.
2. The setting still feels overly simplistic, and many assumptions remain invalidated. For example, is NeuralODE truly the appropriate framework for analyzing such systems? Although prior work has made similar assumptions, to my knowledge these have not been explicitly validated, but rather used as an analytical convenience.
3. Much of the analysis is limited to only two settings, PSD and NSD, which are quite restrictive.
4. It is unclear whether the authors attempted an experiment retraining GPT-2 without layer norms and MLP, and without constraints. While I realize it is too late to request this during the rebuttal, I suspect such a setting might not exhibit the same divergence (or convergence) behavior.

**Overall Assessment:**

In my view, making progress in theoretical ML is extremely challenging, and this work contains elements that could help us better understand the behavior of large language models. Moreover, the authors engaged constructively and thoroughly during the rebuttal, which I greatly appreciate.  Hence, I recommend acceptance.

I recommend that the authors:
1. Revise the paper to address the incorrect proposition identified during the discussion.
2. Consider conducting the experiment described in point 4 above for the revised version.

**Limitations:**

Yes.

**Paper Formatting Concerns:**

No.

**Quality:**

3

**Strengths And Weaknesses:**

**Strengths:**

1. The paper explores an interesting and relevant topic: how token representations evolve as they propagate through the layers of a Transformer, with a focus on whether tokens converge or remain separated.

2. The authors conduct an extensive set of experiments to provide empirical support for their theoretical analysis.

**Weaknesses:**

1. I think the related works section could be improved. It would be helpful if the authors contextualized their contributions more clearly in relation to previous work. Specifically, in lines 37 to 40, the authors state that prior works rely on “unrealistic and impractical assumptions on model parameters... to support the development of richer theoretical results...” and “have not yet offered practical applications or insights...” However, it is not clear what specifically makes those assumptions unrealistic or impractical, nor what assumptions this paper adopts instead or why they are more realistic. Clarifying this distinction would help establish the novelty and practical relevance of the contribution. At present, the difference between this work and earlier studies remains vague, and I could not find any further discussion or elaboration in the appendix.

2. I believe some of the mathematical derivations may contain inaccuracies - please see the questions.

3. I think some of the experiments are constructed in a way that closely reflects the conditions assumed in the theoretical analysis, which may make the results somewhat self-reinforcing. However, it still remains to be seen whether such behaviors occur in real-world settings. Please see questions for a follow-up.

4. The results lack standard deviations.

5. I find the setting to be overly simplistic. While this is not necessarily a flaw, it raises concerns about the practical relevance of the analysis. The Neural ODE model is a coarse approximation of how information propagates through multiple layers of a Transformer. For the analysis, the authors assume that the attention matrix $QK^T$ can be replaced by a fixed matrix $W$, where $Q$ and $K$ are the query and key matrices used in the self-attention mechanism. They further assume for their analysis that $W$ is either positive semidefinite (PSD) or negative semidefinite (NSD). However, in causal attention, $QK^T$ is not symmetric, so $W$ is unlikely to satisfy these conditions. These assumptions simplify the analysis but may significantly limit its applicability to real-world models.

---

> ### Author Rebuttal · Authors · 2025-07-30
>
> We thank the Reviewer for the constructive feedback. Below we address your concerns.
>
> ---
>
> **W1. Difference between this work and earlier studies remains vague.**
>
> **A1.** Let us take this opportunity to clarify the distinction between our results and those in relevant previous works, in order to highlight the novelty and practical relevance of our contributions. Our results directly generalize certain findings from [A] regarding the dynamical properties of tokens in self-attention. In particular, the authors of [A] characterized the convergence, divergence, and clustering behaviors of tokens under the assumption $Q = K = Id$ and $V=Id,-Id$, or parametric. We theoretically generalize these findings in [A] to the case of arbitrary matrices $Q$, $K$, and a matrix $V$ with at least one positive eigenvalue (see Section 3 and Appendix B). We further extend the analysis by investigating the effect of positional encodings in addition to self-attention (see Section 4 and Appendix C). These theoretical results are then empirically verified in practical Transformer settings (see Section 3.4). Moreover, unlike previous works in this line of research, we go beyond theoretical analysis and verification by demonstrating model performance improvements on several tasks (see Section 5 and Appendix D).
>
> We hope that this response clarifies the novelty and originality of our contribution.
>
> **W2-Q6-Q1f. Some of the mathematical derivations may contain inaccuracies regarding the constraint on initial condition. Could the empirical analysis be extended to examine the impact of this constraint?**
>
>
> **A2.** Thank you for pointing out this error. Indeed, the inequality $e^{c\Vert x_\ell(0)\Vert_A^2}-L>0$ is needed to ensure that $\lim_{t \to+\infty}\Vert x_\ell(t)\Vert_A=+\infty$ in the proof of Proposition B3(b). This inequality implies that the initialization and the matrix $A$ must be chosen such that
> $$\Vert x_\ell(0)\Vert_A>\sqrt{\frac{\log(L)}{\max\limits_{u\in\mathbb{R}^D,\Vert u\Vert_A=1}\frac{\Vert u\Vert_W}{\Vert u\Vert_A}}}=O\left(\sqrt{\log(L)}\right).$$
>
> Intuitively speaking, for Proposition B3(b) to hold, the $A$-norm of the initial point must be at least on the order of the square root of the logarithm of the sequence length $L$.
>
> Fortunately, this proposition does not affect any subsequent lemmas or theorems, as Proposition B3(b) is not used later. (Only Proposition B3(a) is referenced.) Our main results (including Theorems 3.1, 3.4 and 3.6) still hold for arbitrary sequence length $L>0$.
>
> **W3. Some of the experiments are constructed in a way that closely reflects the conditions assumed in the theoretical analysis, which may make the results somewhat self-reinforcing**
>
> **A3.** While we include experiments that impose the theoretical constraints described in lines 263–272, none of our empirical validations rely on the NeuralODE assumption of shared weights across layers. Instead, each experiment uses distinct weight matrices per layer to ensure our results hold even when layer parameters vary. Specifically:
>
> (i) Figure 3 shows that our theoretical findings hold without the NeuralODE assumption,
>
> (ii) Figure 4 illustrates clustering behavior in models with FFNs and LNs under the divergence scenario, and
>
> (iii) Figures 6 and 7 show model behavior when relaxing the constraints on $W_{sym}$ or $A_{sym}$, generalizing to the practical pretrained models.
>
> **Q7-Q1ad. Clarification regarding the setup used to produce Figure 6 and extension of experiments in Figure 6 and Section 3.4 to unconstrained settings of pretrained models**
>
> **A4.** To clarify, Figure 6 was generated by removing all LayerNorm and MLP (FFN) modules from a full-architecture pretrained Transformer to isolate pure attention dynamics. We measured token norms and mean pairwise L2 distances across layers under both constrained (divergence, intermediate, convergence) and unconstrained (baseline, similar to real-world Transformers) scenarios.
>
> We visualized relaxed (intermediate/baseline) versus strict (divergence/convergence) constraint regimes. Under intermediate and unconstrained settings, token norms and pairwise L2 distances still diverge, verifying Remarks 3.5, albeit more slowly than in the pure divergence regime. Moreover, Appendix D.5.1’s Figure 7 further shows that, without any constraints on $W_{sym}, A_{sym}$, pretrained GPT-2 token trajectories still exhibit clustering.
>
> **Q1c. Is it possible to evaluate the core Neural ODE assumption?**
>
> **A5.** While the Neural ODE assumption may be violated in real-world settings, our experiments in Section 3.4, our analysis in Section 5.1, and our visualizations in Appendix D.5.1 were conducted without this assumption, thus empirically confirming our theoretical findings in more realistic settings.
>
> **W5-Q1b-Q2. Concerns about simplified setting and practical relevance to causal self-attention. Can one analyze the structure of the matrices $W$ in real models, especially for causal self-attention?**
>
> **A6.** *Regarding the causal attention:* We aim to work with the normal attention, not the causal attention at the moment. The dynamical properties of tokens under causal attention masking represent a different interesting research direction. An initial theoretical result in this area has been presented in [D]. Generalizing this work would certainly be a valuable avenue for future research.
>
> *Regarding the simplified assumptions on the theorem:* Please refer to our answer **A1** to Reviewer p6VJ.
>
> *Regarding the structure of the attention matrices $W = QK^\top$ in real models:* In pretrained practical Transformers, $W$ is not necessarily to be positive or negative definite. However, our analysis is not just restricted to definite matrices. For example, Theorems 3.1 and 3.6 hold for arbitrary $W$, and there are no restrictions on $Q$, $K$, or $V$ in Remark 3.5.
>
> **Q1e. Impact of other architectural components on converge/diverge property**
>
> **A7.** Please refer to our answer **A2** to Reviewer 1jLS.
>
> **W4-Q5. The results lack standard deviations.**
>
> **A8.** We did provide full results (with standard deviation) of Table 1 and 3 in Appendix D.6 (Table 4 and 5, respectively). For Table 2, we provide the full results averaged after 3 runs below:
>
> ||Top-1|Top-5|
> |-|-|-|
> |DeiT baseline|$71.79\pm0.02$|$90.99\pm0.02$|
> |+ convergence|$71.37\pm0.02$|$90.65\pm0.01$|
> |+ intermediate|$71.62\pm0.03$|$90.94\pm0.02$|
> |+ divergence|$71.93\pm0.02$|$91.03\pm0.03$|
>
> **Q3. In the caption of Figure-4 what does it mean by similar shapes?**
>
> **A9.** To clarify, Fig. 4 shows tokens clustered into groups, and tokens within each group propagate through layers in a similar direction.
>
> **Q4. How do you obtain figure 4 and figure 5?**
>
> **A10.** Figure 4 was plotted by projecting onto a manually selected 2D plane (without PCA) to highlight clustering properties. Figure 5 uses tokens that are inherently 2D in our setup, so no dimensionality reduction was needed.
>
> **Q8. Intuition for why divergence leads to improved performance**
>
> **A11.** We claimed in the paper that convergence may negatively affect model performance. Therefore, encouraging the model to avoid the convergence scenario can lead to better practical results. One simple way to avoid convergence is to push the model toward the divergence scenario. The reason convergence may hinder model improvement is that, in this case, token representations tend to shrink toward zero as time $t$ increases. This leads to information loss during training updates, which is not preferable.
>
> **Q9. Practical benefit of Theorems and Propositions 3.1–3.6**
>
> **A12.** The practical benefit of Theorems and Propositions 3.1–3.6 relies on the well-known fact that the continuous-time limit can be used to efficiently approximate modern deep networks. These networks often have a large number of layers, making them well-approximated by continuous-time models. For example, DeepSeek-v3 has approximately 61 layers and can thus be viewed as a discretization of an ODE system. Moreover, the continuous-time limit is a well-established technique in both the machine learning and mathematics communities. It has provided meaningful insights into important behaviors of realistic models, including training dynamics, stability, and generalization.
>
> An alternative way to understand how the described properties evolve layer by layer is through the use of difference equations (instead of differential equations). Exploring this direction is an interesting avenue for future research.
>
> **Answer for Note.** Thank you for the opportunity to address your concerns in detail. Regarding the simplified setting used in our theorems, we adopt a common theoretical approach in which simplified models allow for tractable and rigorous analysis while retaining enough expressive power to capture key patterns seen in practice. Rather than analyzing the exact, highly intricate architectures used in real-world models, theoretical work typically focuses on simplified yet sufficiently general approximations. This methodology aligns with several high-quality works by leading researchers in the field [A, B, C, D]. Fully addressing the complexity of practical Transformer models remains a significant challenge and is currently beyond the scope of comprehensive theoretical analysis.
>
> For a more detailed explanation, please refer to **A6**.
>
> **References**
>
> [A] Geshkovski et al. The emergence of clusters in self-attention dynamics. NeurIPS 2024.
>
> [B] Geshkovski et al. A mathematical perspective on Transformers. Bulletin of the AMS 2024.
>
> [C] Vo et al. Demystifying the Token Dynamics of Deep Selective State Space Models. ICLR 2025 (spotlight).
>
> [D] Karagodin et al. Clustering in Causal Attention Masking. NeurIPS 2024.
>
> -----
>
> We will incorporate the discussion in the revised version. If our responses adequately address the concerns, we kindly hope the evaluation may be reconsidered accordingly. We remain open to further discussion in the next stage of discussion.

---

> > ### Author Response · Authors · 2025-08-04
> > **Follow-Up for Reviewer m1Uq: Update on the Case When W and V are Time Dependent**
> >
> > Thank you once again for your valuable feedback.
> >
> > During the rebuttal process, we recognized that one of the concerns raised by the reviewers relates to the simplifying assumptions in which the attention weights are tied across layers. To address this, we examine whether our theorems can be further generalized to the setting where the model parameters depend on $t$ in our answer A3 to Reviewer waqm. Moreover, by extending the proof of Theorem 3.6--one of the main results that directly supports the experimental findings--we find that the theorem remains valid when both $W$ and $V$ depend on $t$, provided that $V$ has fixed positive eigenvalues for all $t > 0$. We will elaborate on these findings in the revised manuscript.
> >
> > We also wish to clarify that, while several prior works have studied token dynamics in transformers and other deep models, none have addressed the technically challenging scenario in which model parameters vary with time. We view this as an important research direction and plan to explore it in future work.
> >
> > We would appreciate it if you could let us know if our responses have addressed your concerns and whether you still have any other questions about our rebuttal.
> >
> > We are more than happy to engage in follow-up discussions to resolve your concerns, and kindly ask you to consider whether raising your score might better reflect your updated evaluation of our paper.
> >
> > Thank you again for your time and thoughtful comments.
> >
> > Best regards,
> >
> > Authors

---

> > ### Comment · Reviewer_m1Uq · 2025-08-04
> >
> > I thank the authors for their detailed responses and clarifications. I have also gone over the answers to other reviewers. I have some follow-up questions and remarks, listed in the same order as the authors' responses:
> >
> > ---
> >
> > 1. **W2-Q6-Q1f:** I understand that the proofs remain unchanged. However, if I interpret the results correctly, they implicitly require the initialization to be such that the norm of the token embeddings scales with the sequence length. In practice, we typically use standard initialization schemes that are independent of sequence length. This appears to introduce a strong assumption that may not hold in real-world settings. Is there any way to reconcile this discrepancy? To my knowledge, there are no prior results where initialization depends on sequence length, and I suspect such an assumption would be extremely difficult to justify or realize in practice.
> >
> > 2. **W3:**  To clarify my original question, have the transformers in Figures 3 and 4 been initialized in such a way that the PSD or NSD assumptions hold?
> >
> > 3. **Q7-Q1ad:** Would it be possible to investigate the singular value spectrum of intermediate layers in GPT-2? My intuition is that these layers may operate in an intermediate regime (i.e. some of them are positive some are negative. This can also be seen from A2. for reviewer waqm). Additionally, could you clarify the intent behind Figures 7a and 7b? Specifically, how were they generated, what is meant by “clustering” in this context, and how do these results support your overall claims? Furthermore, could you plot inter-token distances across different layers for a few realistic test sequences for this architecture just to see if they converge or diverge? I understand that displaying a figure would not be possible, but a table would suffice.
> >
> > 4. **Q1c:** Just to confirm: has the transformer architecture been modified (as described in lines 263–272) to enforce PSD or NSD properties? If so, how exactly was this done, and what happens if these modifications are removed? For example, is the baseline result in Figure 6 generated without these constraints?
> >
> > 5. **W5-Q1b-Q2:** Apologies, but I’m still unclear, doesn't Remark 3.5 explicitly introduce a constraint?
> >
> > 6. **Q8:** You mention observing clustering in the GPT-2 experiments as well. How do you reconcile this with the intuition that some tokens should naturally cluster while others should diverge (at least in directional terms)? If all tokens are too far apart, attention mechanisms may not function effectively. Could you elaborate?
> >
> > 7. **Q9:** This is more of a comment: I understand that the technique you use is standard in the analysis of gradient descent. Its application to models like transformers, which involve many interacting components, seems less direct and may be difficult to justify in full generality considering the assumptions being made (I also saw the follow-up comment the authors made recently). That said, I see the motivation behind exploring this direction, and I appreciate the inclusion of relevant references.

---

> > > ### Author Response · Authors · 2025-08-05
> > > **(1/3) Response to Reviewer m1Uq**
> > >
> > > Thanks for your further feedback. We answer your additional questions below.
> > >
> > > **W2-Q6-Q1f: I understand that the proofs remain unchanged. However, if I interpret the results correctly, they implicitly require the initialization to be such that the norm of the token embeddings scales with the sequence length. In practice, we typically use standard initialization schemes that are independent of sequence length. This appears to introduce a strong assumption that may not hold in real-world settings. Is there any way to reconcile this discrepancy? To my knowledge, there are no prior results where initialization depends on sequence length, and I suspect such an assumption would be extremely difficult to justify or realize in practice.**
> > >
> > > **A13:** For Proposition 3.3(b), it appears that the assumption that the norm of the tokens scales with the sequence length is unavoidable under the current proof strategy. We conducted simulations with randomly selected parameters that do not satisfy this assumption and found that the differential inequality (11) no longer supports the proof of the divergence of the norms. Therefore, avoiding this assumption would require a different proof approach. In the case where $V$ has a positive eigenvalue, we can follow the proof of Theorem 3.6. We leave the general case for future study.
> > >
> > > However, as mentioned earlier, **Proposition 3.3(b) is an auxiliary proposition and it does not affect the main theorems/results** in our paper (which are Theorems 3.1, 3.4, and 3.6). There is no assumption regarding initialization on these theorems.
> > >
> > >
> > > **W3: To clarify my original question, have the transformers in Figures 3 and 4 been initialized in such a way that the PSD or NSD assumptions hold?**
> > >
> > > **A14:** You are correct that the transformers in Figures 3 and 4 are constrained to satisfy either the PSD or NSD assumptions. These constraints are imposed by reparameterizing the $Q, K, V$ matrices in every attention layers, as detailed in Appendix D.2. However, we intentionally relax the other assumption, namely the NeuralODE constraint, in these experiments. Our goal with these figures is to demonstrate that the theoretical insights still hold in more practical settings where not all theoretical assumptions are strictly enforced. This highlights the relevance of our theory beyond the idealized regime.

---

> > > ### Author Response · Authors · 2025-08-05
> > > **(2/3) Response to Reviewer m1Uq**
> > >
> > > **Q7-Q1ad: Would it be possible to investigate the singular value spectrum of intermediate layers in GPT-2? My intuition is that these layers may operate in an intermediate regime (i.e. some of them are positive some are negative. This can also be seen from A2. for reviewer waqm). Additionally, could you clarify the intent behind Figures 7a and 7b? Specifically, how were they generated, what is meant by “clustering” in this context, and how do these results support your overall claims? Furthermore, could you plot inter-token distances across different layers for a few realistic test sequences for this architecture just to see if they converge or diverge? I understand that displaying a figure would not be possible, but a table would suffice.**
> > >
> > > **A15:** *Regarding the eigenvalue spectrum in GPT-2:* We provide layer-wise statistics of the proportion of positive eigenvalues in $W_{sym}$ and $A_{sym}$:
> > >
> > > | Model | Mean | Std| Min | Max |
> > > |-|-|-|-|-|
> > > | $W_{sym}$ | $46.99$ | $3.43$ | $39.58$ | $51.69$ |
> > > | $A_{sym}$ | $49.97$ | $0.11$ | $49.74$ | $50.13$ |
> > >
> > > As observed, each layer in GPT-2 maintains approximately 50% positive eigenvalues in both matrices $W_{sym}$ and $A_{sym}$, with no layer exhibiting entirely positive or entirely negative eigenvalues. This aligns with your intuition and is consistent with our findings in A2 for reviewer waqm.
> > >
> > > *Regarding the convergence and divergence property in full model:* We measured the distance between tokens and token norms across layers using test sequences from WikiText-103, in both simplified (no FFNs or LayerNorms) and full GPT-2 models.
> > >
> > > In the model omitting FFNs and LayerNorms (Tables 2 and 3), both mean distance between tokens and norm mean grow exponentially across layers, consistent with the behavior of divergence scenario.
> > >
> > > *Table 2: Mean distance betwen tokens (GPT-2 without FFNs and LayerNorms)*
> > >
> > > | # | Input | L1 | L2 | L3 | L4 | L5 | L6 | L7 | L8 | L9 | L10 | L11 | L12 |
> > > |-|-|-|-|-|-|-|-|-|-|-|-|-|-|
> > > |1|4.7e0|9.0e1|1.6e3|2.9e4|4.5e5|1.0e7|1.9e8|6.5e9|1.7e11|3.5e12|7.3e13|1.6e15|7.9e16|
> > > |2|4.7e0|9.5e1|1.9e3|3.4e4|4.9e5|1.0e7|2.2e8|7.8e9|1.8e11|4.2e12|8.3e13|1.8e15|8.2e16|
> > > |3|4.8e0|8.9e1|1.5e3|3.1e4|5.2e5|1.1e7|1.9e8|6.6e9|1.5e11|3.6e12|7.7e13|1.6e15|5.6e16|
> > >
> > > *Table 3: Mean token norm (GPT-2 without FFNs and LayerNorms)*
> > >
> > > | # | Input | L1 | L2 | L3 | L4 | L5 | L6 | L7 | L8 | L9 | L10 | L11 | L12 |
> > > |-|-|-|-|-|-|-|-|-|-|-|-|-|-|
> > > |1|4.8e0|1.6e2|2.2e3|3.5e4|5.2e5|1.1e7|1.9e8|5.6e9|1.3e11|2.7e12|5.4e13|1.2e15|6.1e16|
> > > |2|4.8e0|1.5e2|2.1e3|3.9e4|5.0e5|9.4e6|1.8e8|5.7e9|1.3e11|3.0e12|5.8e13|1.3e15|5.7e16|
> > > |3|4.8e0|1.6e2|1.9e3|4.0e4|5.7e5|1.1e7|1.9e8|5.3e9|1.2e11|2.6e12|5.4e13|1.2e15|4.1e16|
> > >
> > > In contrast, for the full GPT-2 model (which includes FFNs and LayerNorms), we observe that both token norms and distance between tokens increase in the early layers but plateau or decline in the later layers (see Tables 4 and 5). This behavior deviates from our theoretical predictions, indicating that our result does not fully hold in the presence of LayerNorms and FFNs (as discussed in A2 to Reviewer 1jLS). A comprehensive theoretical and empirical analysis of these effects is a promising direction for future work.
> > >
> > >
> > > *Table 4: Mean distance betwen tokens (full GPT-2)*
> > >
> > > | # | Input | L1 | L2 | L3 | L4 | L5 | L6 | L7 | L8 | L9 | L10 | L11 | L12 |
> > > |-|-|-|-|-|-|-|-|-|-|-|-|-|-|
> > > |1|4.69|43.88|59.13|103.43|114.60|124.37|134.71|144.65|157.70|173.58|197.57|230.42|88.51|
> > > |2|4.70|44.85|60.49|104.77|116.07|125.76|135.67|146.69|160.78|177.89|201.93|233.73|74.95|
> > > |3|4.81|46.42|62.57|106.88|117.85|128.07|137.51|148.25|163.11|180.68|204.86|238.15|95.68|
> > >
> > > *Table 5: Mean token norm (full GPT-2)*
> > >
> > > | # | Input | L1 | L2 | L3 | L4 | L5 | L6 | L7 | L8 | L9 | L10 | L11 | L12 |
> > > |-|-|-|-|-|-|-|-|-|-|-|-|-|-|
> > > |1|4.78|52.60|62.60|86.05|93.05|99.76|107.27|116.49|130.89|148.25|180.04|254.02|214.07|
> > > |2|4.80|53.24|63.57|86.93|93.83|100.67|108.51|118.73|132.82|152.33|184.76|258.74|234.81|
> > > |3|4.83|53.85|64.22|88.39|94.89|101.29|107.43|116.81|132.06|150.58|180.75|254.91|224.16|
> > >
> > > ---
> > >
> > > *Regarding clustering property (Figures 7a & 7b):*
> > > By "clustering", we refer to multiple token groups moving in similar directions through layers, not a collapse into one point or complete dispersion. This behavior is observed in:
> > >
> > > - Divergence scenario of simplified models (Figure 3c)
> > > - Divergence scenario of full model with FFNs and LayerNorms (Figure 4)
> > > - Unconstrained full model with FFNs and LayerNorms (Figure 7)
> > >
> > > Unlike the convergence/divergence property of token dynamics, the clustering property can be robustly observed in practical models. Figure 7 is generated by projecting high-dimensional token trajectories onto a manually selected 2D plane. Some tokens are translated (this will not affect their propagation direction) to better visualize directional clustering.

---

> > > ### Author Response · Authors · 2025-08-05
> > > **(3/3) Response to Reviewer m1Uq**
> > >
> > > **Q1c: Just to confirm: has the transformer architecture been modified (as described in lines 263–272) to enforce PSD or NSD properties? If so, how exactly was this done, and what happens if these modifications are removed? For example, is the baseline result in Figure 6 generated without these constraints?**
> > >
> > > **A16:** To enforce the PSD and NSD properties, we derive a reparameterization strategy detailed in Appendix D.2. This method constrains the dynamics by reparameterizing the weight matrices, without modifying the core transformer structure. In Figure 6, the convergence, divergence, and intermediate scenarios are generated using this strategy, whereas the baseline result is obtained using self-attention without any PSD/NSD constraints.
> > >
> > >
> > > **W5-Q1b-Q2: Apologies, but I’m still unclear, doesn't Remark 3.5 explicitly introduce a constraint?**
> > >
> > > **A17:** No, Remark 3.5 does not introduce any constraint. Rather, it suggests that: for every $A$ and $W$
> > > - if they satisfy the assumption $A_{sym}<0$ and $W_{sym}>0$, then all tokens will converge to zero,
> > > - if they do not satisfy this assumption, then all tokens will diverge to infinity.
> > >
> > > **Q8: You mention observing clustering in the GPT-2 experiments as well. How do you reconcile this with the intuition that some tokens should naturally cluster while others should diverge (at least in directional terms)? If all tokens are too far apart, attention mechanisms may not function effectively. Could you elaborate?**
> > >
> > > **A18:** By "clustering", we refer to multiple token groups moving in similar directions through layers, not a collapse into one point or complete dispersion (see Figures 3c, 4, and Appendix Figure 6). In both the divergence scenario and GPT-2, we observe that while tokens diverge overall, local clusters remain coherent enough for attention to compute meaningful similarities. This structured divergence allows intra-cluster attention while reducing inter-cluster overlap. Small clusters (even size 1–2) may exist, reflecting naturally diverging tokens. Thus, our observations are consistent with your intuition: divergence and clustering can coexist to support effective attention.
> > >
> > >
> > > **Q9: This is more of a comment: I understand that the technique you use is standard in the analysis of gradient descent. Its application to models like transformers, which involve many interacting components, seems less direct and may be difficult to justify in full generality considering the assumptions being made (I also saw the follow-up comment the authors made recently). That said, I see the motivation behind exploring this direction, and I appreciate the inclusion of relevant references.**
> > >
> > > **A19:** Thank you for your thoughtful comment and for recognizing the motivation behind our approach. We agree that applying this technique to complex models like transformers, with their many interacting components, is indeed less straightforward than in simpler settings. While our current analysis makes simplifying assumptions, we believe this is an important step toward building more general theoretical tools for understanding such models - a direction we see gaining increasing interest in the community. We will revise the paper to clarify this point and will include additional relevant references to better situate our approach in the broader context.

---

> > > > ### Comment · Reviewer_m1Uq · 2025-08-06
> > > >
> > > > I thank the authors for their comments and appreciate the quick response. I also want to acknowledge the honesty in **A13**, which is commendable and not often seen.
> > > >
> > > > I had a follow-up question regarding **A15**: Which sequences were used to generate the results shown in Table 2 and Table 3? Additionally, over how many sequences were these numbers averaged? More details about the experimental setting would be helpful for interpreting the results.

---

> ### Author Response · Authors · 2025-08-06
> **Response to follow-up question of Reviewer m1Uq**
>
> Thank you again for your quick response. We answer your question below.
>
> **Which sequences were used to generate the results shown in Table 2 and Table 3? Additionally, over how many sequences were these numbers averaged? More details about the experimental setting would be helpful for interpreting the results.**
>
> **A20:** The sequences used to generate the results in Table 2 and Table 3 are taken from the test set of WikiText-103, a language modeling dataset extracted from Wikipedia articles. For each sequence, only the first 100 tokens were used. In Tables 2 and 3, we report results for three randomly selected sequences, each corresponding to one row in the table.
>
> We use a pretrained GPT-2 Small model with 12 Transformer layers, hidden and embedding size of 768, and vocabulary size of 50,257 tokens.
>
> Additionally, we present extended experimental results using a broader range of sequence lengths and a larger set of test sequences to provide a more comprehensive analysis of token behavior. In Tables 6 and 7, we report the mean token norm and the mean distance between tokens for sequence lengths of 16, 64, 100, 128, and 256 tokens. For each length, the reported values are averaged over 100 randomly selected sequences from the test set of WikiText-103.
>
> *Table 6: Mean token norm with varying sequence length (GPT-2 without FFNs and LayerNorms)*
> |Seq len|Input|L1|L2|L3|L4|L5|L6|L7|L8|L9|L10|L11|L12|
> |-|:-:|:-:|:-:|:-:|:-:|:-:|:-:|:-:|:-:|:-:|:-:|:-:|:-:|
> |16|$5.4e0\pm8.2e-2$|$1.7e2\pm3.5e0$|$2.9e3\pm1.3e2$|$3.8e4\pm3.9e3$|$4.1e5\pm2.2e4$|$6.7e6\pm3.7e5$|$1.0e8\pm6.9e6$|$2.4e9\pm3.5e8$|$5.0e10\pm8.6e9$|$9.2e11\pm1.6e11$|$1.8e13\pm3.2e12$|$3.8e14\pm7.2e13$|$1.5e16\pm3.9e15$|
> |64|$4.9e0\pm6.2e-2$|$1.5e2\pm2.0e0$|$2.3e3\pm1.1e2$|$4.1e4\pm6.0e3$|$5.1e5\pm4.2e4$|$8.8e6\pm8.2e5$|$1.5e8\pm1.5e7$|$4.4e9\pm7.8e8$|$1.0e11\pm2.0e10$|$2.1e12\pm4.7e11$|$4.3e13\pm9.4e12$|$9.2e14\pm1.9e14$|$3.6e16\pm8.9e15$|
> |100|$4.8e0\pm5.3e-2$|$1.5e2\pm1.4e0$|$2.1e3\pm1.0e2$|$4.0e4\pm6.6e3$|$5.3e5\pm4.2e4$|$9.8e6\pm8.6e5$|$1.7e8\pm1.9e7$|$5.7e9\pm1.0e9$|$1.3e11\pm2.6e10$|$2.9e12\pm6.1e11$|$5.8e13\pm1.2e13$|$1.2e15\pm2.6e14$|$5.3e16\pm1.2e16$|
> |128|$4.7e0\pm4.8e-2$|$1.5e2\pm1.3e0$|$2.1e3\pm1.1e2$|$4.2e4\pm6.1e3$|$5.5e5\pm4.0e4$|$9.9e6\pm1.0e6$|$1.8e8\pm2.3e7$|$6.0e9\pm1.1e9$|$1.4e11\pm3.2e10$|$3.1e12\pm7.3e11$|$6.4e13\pm1.5e13$|$1.4e15\pm3.2e14$|$6.2e16\pm1.6e16$|
> |256|$4.6e0\pm4.3e-2$|$1.4e2\pm7.4e-1$|$1.9e3\pm8.9e1$|$4.2e4\pm6.6e3$|$5.7e5\pm4.3e4$|$1.0e7\pm1.1e6$|$2.0e8\pm2.6e7$|$7.6e9\pm1.4e9$|$1.9e11\pm3.8e10$|$4.4e12\pm9.7e11$|$9.3e13\pm2.0e13$|$2.1e15\pm4.6e14$|$9.9e16\pm2.7e16$|
>
>
> *Table 7: Mean distance between tokens with varying sequence length (GPT-2 without FFNs and LayerNorms)*
> |Seq len|Input|L1|L2|L3|L4|L5|L6|L7|L8|L9|L10|L11|L12|
> |-|:-:|:-:|:-:|:-:|:-:|:-:|:-:|:-:|:-:|:-:|:-:|:-:|:-:|
> |16|$4.9e0\pm2.0e-1$|$8.1e1\pm7.8e0$|$1.5e3\pm1.2e2$|$2.0e4\pm3.2e3$|$2.7e5\pm5.0e4$|$4.6e6\pm1.0e6$|$7.7e7\pm2.0e7$|$2.1e9\pm6.5e8$|$4.5e10\pm1.4e10$|$8.4e11\pm2.9e11$|$1.5e13\pm5.7e12$|$3.2e14\pm1.2e14$|$1.1e16\pm5.4e15$|
> |64|$4.7e0\pm1.3e-1$|$9.3e1\pm2.9e0$|$1.7e3\pm1.0e2$|$2.9e4\pm3.0e3$|$4.2e5\pm4.6e4$|$8.4e6\pm1.1e6$|$1.6e8\pm2.5e7$|$5.4e9\pm1.1e9$|$1.3e11\pm2.8e10$|$2.8e12\pm6.3e11$|$5.6e13\pm1.2e13$|$1.2e15\pm2.6e14$|$4.6e16\pm1.1e16$|
> |100|$4.7e0\pm1.1e-1$|$9.1e1\pm2.7e0$|$1.6e3\pm1.2e2$|$3.0e4\pm3.5e3$|$4.6e5\pm5.3e4$|$9.8e6\pm1.3e6$|$1.9e8\pm3.2e7$|$7.1e9\pm1.6e9$|$1.7e11\pm3.8e10$|$3.8e12\pm9.0e11$|$7.8e13\pm1.8e13$|$1.7e15\pm3.8e14$|$7.0e16\pm1.7e16$|
> |128|$4.8e0\pm1.0e-1$|$8.9e1\pm2.2e0$|$1.5e3\pm1.1e2$|$3.0e4\pm3.7e3$|$4.5e5\pm6.0e4$|$9.7e6\pm1.5e6$|$1.9e8\pm3.6e7$|$7.4e9\pm1.7e9$|$1.8e11\pm4.5e10$|$4.1e12\pm1.0e12$|$8.5e13\pm2.1e13$|$1.8e15\pm4.7e14$|$8.1e16\pm2.1e16$|
> |256|$5.1e0\pm8.4e-2$|$9.7e1\pm1.0e0$|$1.5e3\pm1.0e2$|$3.1e4\pm3.9e3$|$4.6e5\pm6.3e4$|$1.0e7\pm1.6e6$|$2.2e8\pm4.1e7$|$9.3e9\pm2.1e9$|$2.4e11\pm5.4e10$|$5.8e12\pm1.3e12$|$1.2e14\pm2.8e13$|$2.8e15\pm6.4e14$|$1.2e17\pm3.3e16$|
>
> We also conduct an additional experiment to demonstrate that both the mean norm and the mean pairwise distance between tokens grow exponentially across layers. Specifically, for each choice of sequence length, we fit a linear regression model between the layer depth and the logarithm of the mean norm as well as the logarithm of the mean pairwise distance. The resulting $R^2$ values are shown below:
>
> |Seq len|$R^2$ (log norm)|$R^2$ (log pairwise distance)|
> |-|-|-|
> |16|0.9986|0.9990|
> |64|0.9990|0.9992|
> |100|0.9990|0.9992|
> |128|0.9989|0.9991|
> |256|0.9989|0.9989|
>
> As seen, all $R^2$ values are very close to 1, indicating a strong linear relationship between layer depth and the logarithmic values of the two terms. This confirms that both the mean norm and the mean pairwise distance grow at an exponential rate with depth, which aligns with the expected behavior in the divergence scenario.
>
> Again, we appreciate your feedback and suggestions. We will include these clarification in our revision.

---

> > ### Comment · Reviewer_m1Uq · 2025-08-06
> >
> > I thank the authors for all of their detailed comments and clarifications. I have no further questions. I will update my score accordingly based on the new results and the clarifications provided. Thank you once again for the responses.

---

> > > ### Author Response · Authors · 2025-08-06
> > > **Thanks for your endorsement!**
> > >
> > > Thanks for your response, and we appreciate your endorsement.

---

> > > ### Author Response · Authors · 2025-08-09
> > > **Again, thanks for your endorsement!**
> > >
> > > Dear Reviewer m1Uq,
> > >
> > > Thank you once again for your time and for providing such thoughtful feedback on our work. We would be grateful if you could let us know whether you have any last remaining questions or concerns regarding our rebuttal before the discussion period ends. We would be pleased to engage in further discussion to address any outstanding points.
> > >
> > > With sincere appreciation and best regards,
> > >
> > > Authors

---

### Official Review · Reviewer_p6VJ · 2025-07-01

**Clarity:** 3
**Significance:** 2
**Originality:** 2
**Rating:** 4
**Confidence:** 3

**Summary:**

This manuscript takes a dynamical-systems viewpoint—modeling layers of an attention-only deep neural network as continuous—with the motivation of exploring the evolution of token embeddings through the layers of a transformer during pretraining.  The authors consider a regime where weight matrices are tied (i.e., identical) across layers.  In this regime, they find that activations from different inputs either converge or diverge through the layers, depending on whether the symmetric components of certain matrices ($W = Q K^\top$ and $A =  Q K^\top (V^{-1})^\top$) are positive definite (i.e., have only positive eigenvalues) or negative definite (i.e., have only negative eigenvalues).  Effects of positional embeddings are considered.  The theoretical predictions are borne out by transformers that are pretrained with weight matrices tied across layers, even when the transformers include FFN layers in addition to the attention layers.  In the end, the paper suggests ways to promote divergent embeddings.  Appendices provide derivations of all claims.

**Questions:**

The authors assert that their first contribution is to “provide conditions that predict whether tokens in self-attention converge to zero or diverge to infinity”.  By standard use of these terms, this claim is inscrutable.  I am guessing the authors mean that the magnitude of token embeddings in a transformer are converging or diverging?  Can the authors provide citation(s) of this issue in real transformers?

Regarding the framework laid out in Sec. 2.2, the authors may want to consider causal attention masking in future work, since this would be most relevant for autoregressive transformers.

It is not clear how the eigenvalues of $A$ correspond to the eigenvalues of its symmetrization $A_\text{sym}$.  In the general case, I’m not aware of any easy correspondence.  Accordingly it doesn’t seem to be the case that, e.g., the positive-definiteness of one implies the positive-definiteness of the other.  In the paper, sometimes positive-definiteness is talked about in terms of $A$ and sometimes in terms of $A_\text{sym}$, and sometimes in terms of $A$ after assuming it is symmetric such that $A = A_\text{sym}$.  How should the reader think about these relationships?  In particular, the claims starting on Line 128 involving the positive definiteness of $A$ don’t appear to assume that $A$ is symmetric, but then Thm. 1 assumes that $A$ is a symmetric matrix and Remark 3.2 is about the positive definiteness of the symmetrization of $A$.


Minor suggestions:

Line 41 and Section A:
“Only a limited number of studies in the literature provide a theoretical understanding of the internal structure of learned representations in pre-trained Transformer models (see Section A for a comprehensive list of relevant works).”
This field moves fast, so it seems overly confident to suggest a “comprehensive” list of relevant works that provide understanding of the internal structure of learned representations.  I can certainly think of many more.  Perhaps remove the word “comprehensive”.

Line 48:
It seems that “fourth-fold” should be changed to “fourfold”.

Line 118:
Consider explicitly stating the intended meaning/definition of “equivariant” in the statement that “it is equivariant to the standard Euclidean norm”

Line 157:
Typo:  Change “randomly chosen of model parameters” to “randomly chosen model parameters”

**Ethical Concerns:**

["NO or VERY MINOR ethics concerns only"]

**Final Justification:**

I have slightly upped my recommendation since the authors have been able to show relevance of their work when the (otherwise quite restrictive) weight-tying assumption is at least slightly relaxed.

**Limitations:**

yes

**Paper Formatting Concerns:**

no such concerns

**Quality:**

3

**Strengths And Weaknesses:**

This paper does a good job of stating the equations and assumptions for their modeling framework.  Once this framework is set up, the ensuing math and dynamical-systems analysis all seems valid.  (Good Clarity and fine Quality for the topic of study.)

The primary weakness of this paper is that the modeling assumptions seem to create a serious disconnect from real transformers, such that the predictions seem irrelevant.  The form of attention equations looks similar, but the continuous-time limit has no obvious motivation other than it allows for a dynamical system that can be analyzed.  Tying the attention weight matrices across layers of a transformer is a strong restriction, which mirrors the ALBERT architecture but does not reflect standard architectures of interest.  As a further significant restriction, this paper does not incorporate any analog of the FFN layers of transformers in the theoretical analysis.  I recognize that other papers like Ref. [17] have been published before using a similar modeling framework, but I don’t believe this precedent provides sufficient justification for the importance of the current analysis for understanding real neural nets which would be an expected standard for a conference like NeurIPS.  Rather, the existence of several other papers that make very similar assumptions reduces the novelty of this approach and makes the current work feel quite iterative.  (Low Originality and Significance.)  I believe this paper would be better suited for a dynamical systems audience looking for neat dynamics, rather than an audience trying to understand transformers.

The authors say that “Our results are also validated against pre-trained Transformer models, confirming that
the theoretical insights carry over to real-world settings.”  The authors indeed show some relevance of their theoretical predictions for the transformers they train, including those with FFN layers in addition to attention layers, but my understanding is that these transformers have their weight matrices tied across layers, which is not much of a real-world setting (besides the ALBERT architecture).

Some standard ML terms seem to be used unusually in this paper. In particular, the word “token” is used throughout the paper to rather mean something like “token embedding” or “residual stream”.  The stated study of how tokens evolve is really about how the activations in the residual stream evolve through the layers.  For example, “when tokens move closer” would be more precise if it was changed to say “when embeddings move closer in deeper layers of the network”.  It would be much more conventional for “tokens” to refer to the symbols from a fixed discrete set, rather than real values.

---

> ### Author Rebuttal · Authors · 2025-07-30
>
> We thank the Reviewer for the constructive feedback. Below we address your concerns.
>
> -----
>
> **W1. Concerns include the reliance on a continuous‐time limit and simplifying assumptions, specifically tying attention weights across layers and omitting feed‐forward and layer‐normalization modules, which undermine both the work’s relevance to real Transformer architectures and its originality.**
>
> **A1.** *Regarding the use of continuous-time limit*: We respectfully disagree with the reviewer about the comment "continuous-time limit has no obvious motivation other than it allows for a dynamical system that can be analyzed". Modern deep networks often have a large number of layers, making them well-approximated by continuous-time models. For example, ResNets can be seen as discretizations of ODEs, and this perspective becomes increasingly accurate as depth increases. Moreover, the continuous-time limit is a well-established technique in both the machine learning and mathematics communities. It has led to important insights into training dynamics, stability, and generalization, and underpins models like Neural ODEs, State Space Models, and Liquid Neural Networks. Thus, it is both theoretically justified and practically relevant.
>
> *Regarding your concern about our simplifying assumptions, namely, tying the attention weight matrices across layers and excluding the FFN/LayerNorm*: While your concern is valid, it is common for theorists to adopt simplified settings of the model to enable feasible theoretical analysis, while still retaining sufficient capacity to capture key patterns observed in practical scenarios. Several high-standard related works by notable researchers also operate within such simplified settings, for example:
>
> [A] Geshkovski et al. The emergence of clusters in self-attention dynamics. NeurIPS 2024. (The assumption $Q=K=V=Id$ was used.)
>
> [B] Geshkovski et al. A mathematical perspective on Transformers. Bulletin of the AMS 2024. (The assumption $Q=K=V=Id$ was used.)
>
> [C] Vo et al. Demystifying the Token Dynamics of Deep Selective State Space Models. ICLR 2025 (spotlight). (The assumption dim $=1$ and time-invariant parameters were used.)
>
> Our results generalize certain findings in [A] by employing more general assumptions on $Q, K, V$, and by additionally analyzing the effect of positional encodings. We believe that this generalization is already nontrivial.
>
> The general case involving a full and practical Transformer model remains highly challenging, and to the best of our knowledge, no existing work has achieved this result to date.
>
>
>
>
>
> **W2. The authors say that “Our results are also validated against pre-trained Transformer models, confirming that the theoretical insights carry over to real-world settings.” The authors indeed show some relevance of their theoretical predictions for the transformers they train, including those with FFN layers in addition to attention layers, but my understanding is that these transformers have their weight matrices tied across layers, which is not much of a real-world setting (besides the ALBERT architecture).**
>
>
> **A2.**  To clarify, we did verify our theoretical results using different settings of practical Transformer architectures in Sections 3.4 and 5.1, none of which restrict weights from being shared across layers. In particular:
> (i) Figure 3 demonstrates that our theoretical findings hold in the absence of the weight-tying assumption,
> (ii) Figure 4 illustrates the clustering behavior of models with FFNs and LNs, but constrained to the divergence scenario, and
> (iii) Figures 6 and 7 demonstrate the behavior of the models when lifting the constraints on  $W_{sym}$ or $A_{sym}$, generalizing to the practical pre-trained models.
>
>
> **W3-Q1. Some standard ML terms seem to be used unusually in this paper. In particular, the word “token” is used throughout the paper to rather mean something like “token embedding” or “residual stream”. The stated study of how tokens evolve is really about how the activations in the residual stream evolve through the layers. For example, “when tokens move closer” would be more precise if it was changed to say “when embeddings move closer in deeper layers of the network”. It would be much more conventional for “tokens” to refer to the symbols from a fixed discrete set, rather than real values (W3). The authors assert that their first contribution is to “provide conditions that predict whether tokens in self-attention converge to zero or diverge to infinity”. By standard use of these terms, this claim is inscrutable. I am guessing the authors mean that the magnitude of token embeddings in a transformer are converging or diverging? Can the authors provide citation(s) of this issue in real transformers? (Q1)**
>
>
> **A3.**  The way we use the term "token" in this paper is consistent with its usage in many previous works in this line of research. For example, it appears in high-standard publications [A, B, C, D]. In particular, the authors of [A] characterized the convergence/divergence properties of tokens when $Q=K=V=Id$. Our results generalize the findings in [A] by employing more general assumptions on $Q,K,V$, and by also analyzing the effect of positional encodings.
>
> Following your suggestion, we will add a remark to explain alternative terms for "token" used in other areas, such as "token embedding" or "residual stream."
>
>
>
>
> **Q2: Regarding the framework laid out in Sec. 2.2, the authors may want to consider causal attention masking in future work, since this would be most relevant for autoregressive transformers.**
>
> **A4.**  The dynamical properties of tokens under causal attention masking represent an interesting research direction. An initial theoretical result in this area was presented in [D]. Generalizing this work would certainly be a valuable avenue for future research. We will add this to the Future Work section of our paper.
>
> [D] Karagodin et al. Clustering in Causal Attention Masking. NeurIPS 2024.
>
> **Q3. It is not clear how the eigenvalues of $A$ correspond to the eigenvalues of its symmetrization $A_{sym}$. In the general case, I’m not aware of any easy correspondence. Accordingly it doesn’t seem to be the case that, e.g., the positive-definiteness of one implies the positive-definiteness of the other. In the paper, sometimes positive-definiteness is talked about in terms of $A$ and sometimes in terms of $A_{sym}$, and sometimes in terms of $A$ after assuming it is symmetric such that $A=A_{sym}$. How should the reader think about these relationships? In particular, the claims starting on Line 128 involving the positive definiteness of $A$ don’t appear to assume that $A$ is symmetric, but then Thm. 1 assumes that $A$ is a symmetric matrix and Remark 3.2 is about the positive definiteness of the symmetrization of $A$.**
>
>
> **A5.** We did not claim any correspondence between the eigenvalues of $A$ and those of $A_{sym}$. Instead, we used the following two distinct assumptions:
>
> (a) $A$ is positive definite;
>
> (b) $A_{sym}$ is positive definite.
>
> When we use (a), this means that $A = A_{sym}$ and is positive definite. However, when we use (b), $A$ is not necessarily symmetric, but its symmetric part $A_{sym}$ must be positive definite. To clarify, (a) implies (b), but (b) does not imply (a).
>
>
>
> **MS1. Line 41 and Section A: “Only a limited number of studies in the literature provide a theoretical understanding of the internal structure of learned representations in pre-trained Transformer models (see Section A for a comprehensive list of relevant works).” This field moves fast, so it seems overly confident to suggest a “comprehensive” list of relevant works that provide understanding of the internal structure of learned representations. I can certainly think of many more. Perhaps remove the word “comprehensive”.**
>
> **A6.** We agree to remove the word "comprehensive" in the revised version. In addition, we will add further relevant works to the related works section.
>
> **MS2. Line 48: It seems that “fourth-fold” should be changed to “fourfold”.**
>
> **A7.** We will change “fourth-fold” to “fourfold” in the revised version.
>
>
> **MS3. Line 118: Consider explicitly stating the intended meaning/definition of “equivariant” in the statement that “it is equivariant to the standard Euclidean norm”**
>
> **A8.** The correct statement is: “it is equivalent to the standard Euclidean norm.” Two norms on $\mathbb{R}^D$ are equivalent if and only if they induce the same topology. It is well known that all norms on $\mathbb{R}^D$ are equivalent. We will update this in the revised version.
>
>
> **MS4. Line 157: Typo: Change “randomly chosen of model parameters” to “randomly chosen model parameters”**
>
> **A9.** We will update this in the revised version.
>
> -----
>
> We will incorporate the discussion in the revised version. If our responses adequately address the concerns, we kindly hope the evaluation may be reconsidered accordingly. We remain open to further discussion in the next stage of discussion.

---

> > ### Author Response · Authors · 2025-08-04
> > **Follow-Up for Reviewer p6VJ: Update on the Case When W and V are Time Dependent**
> >
> > Thank you once again for your valuable feedback.
> >
> > During the rebuttal process, we recognized that one of the concerns raised by the reviewers relates to the simplifying assumptions in which the attention weights are tied across layers. To address this, we examine whether our theorems can be further generalized to the setting where the model parameters depend on $t$ in our answer A3 to Reviewer waqm. Moreover, by extending the proof of Theorem 3.6--one of the main results that directly supports the experimental findings--we find that the theorem remains valid when both $W$ and $V$ depend on $t$, provided that $V$ has fixed positive eigenvalues for all $t > 0$. We will elaborate on these findings in the revised manuscript.
> >
> > We also wish to clarify that, while several prior works have studied token dynamics in transformers and other deep models, none have addressed the technically challenging scenario in which model parameters vary with time. We view this as an important research direction and plan to explore it in future work.
> >
> > We would appreciate it if you could let us know if our responses have addressed your concerns and whether you still have any other questions about our rebuttal.
> >
> > We are more than happy to engage in follow-up discussions to resolve your concerns, and kindly ask you to consider whether raising your score might better reflect your updated evaluation of our paper.
> >
> > Thank you again for your time and thoughtful comments.
> >
> > Best regards,
> >
> > Authors

---

> > ### Comment · Reviewer_p6VJ · 2025-08-07
> > **Reply to authors' rebuttal**
> >
> > I thank the authors for their detailed response.
> >
> > > (i) Figure 3 demonstrates that our theoretical findings hold in the absence of the weight-tying assumption
> >
> > Great!
> > Does the paper claim this?  If not, maybe it should.
> > I'm confused how this can be consistent with what is written in the manuscript though, since the panels of Fig. 3 talk about properties of the A matrix and the W matrix for the network.  If not tied, there would be many A and W matrices.  So what is going on?  (I'm guessing something like "this property is true for every one of the A and W matrices".)  I suggest the authors are more explicit about this both when talking about Fig. 3 and in the relevant appendix.

---

> > > ### Author Response · Authors · 2025-08-07
> > > **Reply to reviewer p6VJ**
> > >
> > > Thank you for your response. We answer your question below.
> > >
> > > **"Figure 3 demonstrates that our theoretical findings hold in the absence of the weight-tying assumption." Does the paper claim this? If not, maybe it should.**
> > >
> > > **Answer:** Thank you for your suggestion. We agree that this clarification is important for improving the clarity of the paper. We will revise the manuscript to more clearly highlight our contribution: specifically, that we have conducted experiments demonstrating our theoretical findings remain valid even without the weight-tying assumption.
> > >
> > > **I'm confused how this can be consistent with what is written in the manuscript though, since the panels of Fig. 3 talk about properties of the A matrix and the W matrix for the network. If not tied, there would be many A and W matrices. So what is going on? (I'm guessing something like "this property is true for every one of the A and W matrices".)**
> > >
> > > **Answer:**  Yes, you’ve understood it correctly. In Figure 3, we illustrate the token dynamics across layers in a pretrained Transformer. The matrices $A$ and $W$ are not shared between layers; each layer has its own distinct $A$ and $W$. However, all layers follow the same constraint pattern-either in the convergence scenario ($A_{\text{sym}} \prec 0$ and $W_{\text{sym}} \succ 0$) or in the divergence scenario ($A_{\text{sym}} \succ 0$ and $W_{\text{sym}} \succ 0$). We will include this clarification in the revised version of the paper.
> > >
> > > We hope our responses have successfully addressed your concerns. We would be happy to provide any additional clarifications if needed. Please let us know if you have any remaining questions or concerns.
> > >
> > > Thank you again for your time and valuable feedback.

---

### Official Review · Reviewer_1jLS · 2025-07-01

**Clarity:** 4
**Significance:** 4
**Originality:** 4
**Rating:** 5
**Confidence:** 5

**Summary:**

This is a very good paper. It extends the existing line of work on studying signal propagation in Transformer models, where the parameters are fixed (pre-trained model) and the evolution of tokens across layers is analyzed. The authors take a « neural ODE » point of view in which time plays the role of a layer, and this perspective is well-accepted in the community as being particularly helpful in doing theory. This line of research is clearly fundamental and critical for improving the interpretability and understanding of how large language models work.

**Questions:**

1. I have a question regarding the addition of positional encoding in the equation. I thought positional encoding is added to the initial condition, and that’s it. Why does it appear in the ODE in equation (5)? I am not totally sure I understand this point.

2. The authors are also very welcome to comment on the possible impact that an MLP layer (or a normalisation layer) may have on the divergence, or the avoidance thereof.

3. Moreover, there is a paper by Koubbi, Boussard and Hernandez, where they also generalise the results of Geshkovski et al. to somewhat more general parameters, but it’s not cited herein. Perhaps the authors can discuss the differences with that paper.

**Ethical Concerns:**

["NO or VERY MINOR ethics concerns only"]

**Final Justification:**

I am satisfied by the answers of the authors, and my issues have been resolved.

**Limitations:**

yes

**Quality:**

3

**Strengths And Weaknesses:**

In particular, the authors address the effect of positional encoding on the dynamics of tokens within the Transformer architecture, which has not been seriously considered before. The novelty is clear. There are also clear extensions on the existing results of Transformer dynamics to more general classes of parameter matrices (key-query and value matrices, to be more precise).

---

> ### Author Rebuttal · Authors · 2025-07-30
>
> We thank the Reviewer for the constructive feedback. Below we address your concerns.
>
> -----
>
>
> **Q1. I have a question regarding the addition of positional encoding in the equation. I thought positional encoding is added to the initial condition, and that’s it. Why does it appear in the ODE in equation (5)? I am not totally sure I understand this point.**
>
> **A1.** Indeed, adding the positions $p_i$ to the initial conditions is equivalent to fixing the initial condition and adding $p_i$ to $x_i$, as in Equation (5).
>
> **Q2. The authors are also very welcome to comment on the possible impact that an MLP layer (or a normalisation layer) may have on the divergence, or the avoidance thereof.**
>
> **A2.** Unfortunately, not all theoretical results continue to hold when an MLP or LayerNorm is introduced. However, under the divergence scenario where MLP/LayerNorm are present, Figure 4 of Section 3.4 and Figure 7 of Appendix D.5.1 in our manuscript show that tokens still form distinct clusters, with each group following a similar trajectory pattern. This suggests that our observation on directional token clustering persists even in a fully structured Transformer model with MLP/LayerNorm. A theoretical analysis for these cases requires further adaptation, which we leave for future research.
>
> **Q3. Moreover, there is a paper by Koubbi, Boussard and Hernandez, where they also generalise the results of Geshkovski et al. to somewhat more general parameters, but it’s not cited herein. Perhaps the authors can discuss the differences with that paper.**
>
> **A3.**  Thank you for pointing out the interesting work in [A]. As far as we are aware, the authors of [A] assume that the triple $(Q, K, V)$ already leads to a clustering phenomenon (at infinity) of tokens and investigate how such clustering is affected by small perturbations of the triple through the addition of low-rank components. In contrast to this approach, we directly characterize the dynamical properties of tokens for the triple $(Q, K, V)$ based on an analysis of their eigenvalues. Furthermore, we analyze the effect of positional encoding on the dynamics. Beyond that, we also propose model modifications to improve performance across different tasks. We will add reference [A] and include this discussion in the revised version of the paper.
>
>
> [A] Koubbi, Boussard and Hernandez et al., The Impact of LoRA on the Emergence of Clusters in Transformers, arxiv.
>
> -----
>
> We will incorporate the discussion in the revised version. We remain open to further discussion in the next stage of discussion.

---

> > ### Comment · Reviewer_1jLS · 2025-08-05
> >
> > Totally fine wrt A2 and A3.
> > I am not sure I understand A1 as of yet. Are the authors considering the equation for the pure tokens (w/o Positional encoding), and then view the positional encoding as a "source term" in the equation? In this case, the equation is permutation invariant, is it not?

---

> ### Author Response · Authors · 2025-08-05
> **Response to Reviewer 1jLS**
>
> We thank the reviewer again for your insightful reviews and valuable feedback. We answer your additional question below.
>
> **Q1: I am not sure I understand A1 as of yet. Are the authors considering the equation for the pure tokens (w/o Positional encoding), and then view the positional encoding as a "source term" in the equation? In this case, the equation is permutation invariant, is it not?**
>
> **A4:** Let us take this opportunity to explain why *adding the positions $p_i$ to the initial conditions*, as you mentioned, is equivalent to *fixing the initial condition and adding $p_i$ to $x_i$*, as in Equation (5). Indeed, from Equation (5), if we consider the transformation $y_l(t) = x_l(t) + p_l$ for all $l = 1, \ldots, L$, then the initial condition corresponding to $y_l$ becomes
> $$y_l(0) = x_l(0) + p_l,$$
> while the differential equations governing $y_l(t)$ are exactly those of the sole attention dynamics. Because of this, positional encoding is added to the initial condition (if we follow the dynamics of $y_l$), as you mentioned, and this is equivalent to fixing the initial condition and adding $p_i$ to $x_i$, as in Equation (5) of our paper.

---

> > ### Comment · Reviewer_1jLS · 2025-08-06
> >
> > Thanks for the clarification.

---

> > > ### Author Response · Authors · 2025-08-06
> > > **Thanks for your endorsement!**
> > >
> > > Thanks for your response, and we appreciate your endorsement.

---

### Official Review · Reviewer_waqm · 2025-07-01

**Clarity:** 3
**Significance:** 2
**Originality:** 3
**Rating:** 4
**Confidence:** 3

**Summary:**

This paper studies the dynamics of tokens in transformers (removing all blocks other than the self-attention) when all the parameters are trained and fixed. By viewing the forward of tokens as a continuous (partial) differential equation, the authors are able to reveal the divergence ($||x|| \to \infty$) and convergence ($x \to 0$) phases of tokens that depend explicitly on $W$ (by merging the query matrix $Q$ and key matrix $K$) and the value matrix $V$. As an application, by exploring how different positional encodings can change the dynamics, the authors propose a new simple technique to encourage token divergence to improve the performance of transformers.

**Questions:**

1. Could the authors explain what the conclusions of the current manuscript will be if all the weights depend on the variable $t$?

2. Could the conclusions/prediction be applied to real-world pre-trained models? And if so, what will the comparison between a randomly initialized model and the pre-trained model be regarding the conditions given by this paper?

**Ethical Concerns:**

["NO or VERY MINOR ethics concerns only"]

**Final Justification:**

The authors have addressed my concerns regarding the conditions of the initialized model paramters, while the setting regarding time-dependent parameters cannot be solved in the current manuscript. As this is a follow-up work, I maintain my score of 4.

**Limitations:**

yes

**Paper Formatting Concerns:**

There is no formatting issue.

**Quality:**

3

**Strengths And Weaknesses:**

## Strengths

1. This paper is overall well written. In particular, the authors provide a clear motivation for the problem being studied and organize the flow of the paper very well, hence it is very easy to understand the main results, i.e., the main results for the dynamics of tokens are clearly summarized as three parts: the distance between tokens, the convergence of tokens, and the divergence of tokens.

2. The conditions for the divergence and convergence phases of tokens are surprisingly simple: by examining whether $W$ and $A$ are positive definite or not. Thus the conditions might be easily checked in practice by examining the eigenvalues of these matrices.

## Weaknesses

1. Please correct me if I understand the setting in a wrong way: the parameters are indeed trained and fixed, however, the time variable $t$ refers to the depth of the transformers. Hence the fact that $W$ and $A$ do not depend on $t$ means that the merged weight $W$ of attention and the value matrix $V$ are also the same for all layers. This does not align with the actual case. As the cluster of tokens has already been studied by prior works, I think it is better to take a step further to remove the limitation by considering a case where $W$ can depend on $t$.

2. One aspect that has not been touched by this paper is the statistics of the conditions for $W$ and $A$ of real-world pre-trained models (not manually designed by some re-parameterization as in Appendix D.2). Specifically, the condition that the value matrix $V$ is invertible cannot necessarily be guaranteed in practice. Thus it is unclear whether these pre-trained models indeed exhibit such divergence and convergence of tokens that can match the conclusion of this paper. In a word, it is not clear whether these conditions and the corresponding divergence/convergence of tokens only exist in the special setting of the paper or are more general across a wide range of settings.

---

> ### Author Rebuttal · Authors · 2025-07-30
>
> We thank the Reviewer for the constructive feedback. Below we address your concerns.
>
> -----
>
> **W1. Please correct me if I understand the setting in a wrong way: the parameters are indeed trained and fixed, however, the time variable $t$ refers to the depth of the transformers. Hence the fact that $W$ and $A$ do not depend on $t$ means that the merged weight $W$ of attention and the value matrix $V$ are also the same for all layers. This does not align with the actual case. As the cluster of tokens has already been studied by prior works, I think it is better to take a step further to remove the limitation by considering a case where $W$ can depend on $t$.**
>
> **A1.** Yes, you are correct. In our theorems, we assume that $W$ and $A$ are independent of $t$. We empirically verify our theoretical findings in real-world pretrained transformers in Section 3.4.
>
> While your concern regarding the limited scope of practical scenarios under these assumptions is valid, it is common for theorists to adopt simplified settings of the model to enable feasible theoretical analysis, while still retaining sufficient capacity to capture key patterns observed in practical scenarios. Several high-standard related works by notable researchers also operate within such simplified settings, for example:
>
> [A] Geshkovski et al. The Emergence of Clusters in Self-Attention Dynamics. NeurIPS 2024. (The assumption $Q=K=V=Id$ was used.)
>
> [B] Geshkovski et al. A Mathematical Perspective on Transformers. Bulletin of the AMS 2024. (The assumption $Q=K=V=Id$ was used.)
>
> [C] Vo et al. Demystifying the Token Dynamics of Deep Selective State Space Models. ICLR 2025 (spotlight). (The assumptions dim $=1$ and time-invariant parameters were used.)
>
> Our results generalize certain findings in [A] by employing more general assumptions on $Q$, $K$, and $V$, and even analyze the effect of positional encodings. We believe that this generalization is already nontrivial.
>
> The case where $W$ and $A$ depend on $t$ remains highly challenging, and to the best of our knowledge, no existing work has achieved this result so far. We leave this interesting problem for future work.
>
>
>
> **W2-Q2. One aspect that has not been touched by this paper is the statistics of the conditions for $W$ and $A$ of real-world pre-trained models (not manually designed by some re-parameterization as in Appendix D.2). Specifically, the condition that the value matrix $V$ is invertible cannot necessarily be guaranteed in practice. Thus it is unclear whether these pre-trained models indeed exhibit such divergence and convergence of tokens that can match the conclusion of this paper. In a word, it is not clear whether these conditions and the corresponding divergence/convergence of tokens only exist in the special setting of the paper or are more general across a wide range of settings (W2). Could the conclusions/prediction be applied to real-world pre-trained models? And if so, what will the comparison between a randomly initialized model and the pre-trained model be regarding the conditions given by this paper? (Q2)**
>
> **A2.** We performed an empirical analysis of real-world pretrained Transformer models to verify the conditions in our theory. Specifically, for each model, we measured the mean percentage across all layers of (i) “near-zero” eigenvalues of the value matrix $V$ (using threshold $\epsilon = 10^{-3}$), and (ii) positive eigenvalues of the symmetrized matrices $W_{\mathrm{sym}}$ and $A_{\mathrm{sym}}$.
>
> Table 1 reports the mean ± standard deviation of these percentages for GPT-2 XL, DistilGPT2, and LLaMA-2 13B. We observe that: (i) In all cases, $V$ exhibited 0% near-zero eigenvalues, indicating that $V$ is invertible in practice. (ii) The proportion of positive eigenvalues in both $W_{\mathrm{sym}}$ and $A_{\mathrm{sym}}$ is approximately 50%, which matches the divergence regime predicted by Remark 3.5 if omitting FFNs and LayerNorms. The divergence behaviour of an unconstrained/real-world pretrained model is also illustrated as 'baseline' case of Figure 6 in Section 5.1.
>
> These findings align with theoretical expectations: (i) the set of singular matrices has measure zero in continuous parameter spaces, and (ii) unconstrained weight matrices, whether randomly initialized or learned, tend to have eigenvalue distributions symmetric about zero. We also repeated this analysis on a randomly initialized model (GPT-2 XL reinitialized) and obtained similar results (approximately 50% positive eigenvalues in $W_{\mathrm{sym}}$, $A_{\mathrm{sym}}$, and no near-zero eigenvalues in $V$).
>
>
> *Table 1: Mean ± std. dev. of eigenvalue statistics (in %) across layers.*
> |Model|% positive eigenvalue of $W_{sym}$|% positive eigenvalue of $A_{sym}$|% near-zero-eig of $V$|
> |-|-|-|-|
> |GPT2-xl|$50.20\pm3.76$|$49.99\pm0.06$|$0.00\pm0.00$|
> |DistilGPT2|$46.01\pm4.36$|$49.96\pm0.16$|$0.00\pm0.00$|
> |Llama2 13B|$52.35\pm2.84$|$50.00\pm0.02$|$0.00\pm0.00$|
> |GPT2-xl random reinitialized|$50.20\pm2.85$|$49.99\pm0.06$|$0.00\pm0.00$|
>
> **Q1. Could the authors explain what the conclusions of the current manuscript will be if all the weights depend on the variable $t$?**
>
> **A3.** Unfortunately, the theorems may no longer hold if the weights depend on the variable $t$. However, our simulations using randomly selected time-dependent weights suggest that the theorems may still hold under analogous assumptions. For example: Theorem 3.1 appears to remain valid if $A$ is definite and maintains the same sign for all $t > 0$; Theorem 3.4 still holds if $A < 0$ and $W > 0$ for all $t > 0$; and Theorem 3.5 continues to hold if $V$ possesses a fixed positive eigenvalue for all $t > 0$. We will include this discussion in the revised version of the paper.
>
> -----
>
> We will incorporate the discussion in the revised version. If our responses adequately address the concerns, we kindly hope the evaluation may be reconsidered accordingly. We remain open to further discussion in the next stage of discussion.

---

> > ### Comment · Reviewer_waqm · 2025-08-03
> >
> > Thankse for your detailed response.
> >
> > Although I understand that studying simplified settings is acceptable, I believe it would be better to generalize settings in prior works further, e.g., by considering a $W$ depending on $t$ as in my original comment, to align with the practical setting, as there are already a lot of works in this direction and, as a comparison, the conclusions achieved by this paper are not significantly new.
> >
> > My other concerns have been resolved.

---

> > > ### Author Response · Authors · 2025-08-03
> > > **Update on the case when $W$ and $V$ are time-dependent**
> > >
> > > Thanks for your further feedback, and we appreciate your endorsement.
> > >
> > > Following your suggestion, we have verified whether our theorems can be further generalized to the setting where the model parameters depend on $t$ in A3 in our rebuttal above. Furthermore, by following the lines of the proof of Theorem 3.6 - one of the main results that directly supports the experimental findings - we have found that **Theorem 3.6 remains valid when both $W$ and $V$ depend on $t$, provided that $V$ has a fixed positive eigenvalue for all $t > 0$**. We will elaborate on this in the revised manuscript.
> > >
> > > Additionally, we would like to clarify that although several prior works have investigated token dynamics in transformers and other deep models, none have addressed the technically challenging scenario where model parameters are time-dependent. We recognize this as a valuable direction and plan to explore it in future work.

---

### Author Response · Authors · 2025-08-09
**Thank You to the Chairs and Reviewers!**

Dear Chairs and Reviewers,

We sincerely thank you for your thoughtful and constructive feedback throughout the review and discussion phases. We will incorporate the additional results and suggested clarifications during the rebuttal and discussions with reviewers into our revised manuscript.

Once again, we greatly appreciate your time and valuable input.

Best regards,

Authors

---

### Note · Authors · 2025-08-13

Dear AC and Reviewers,

Thanks for your thoughtful reviews and valuable comments, which have helped us improve the paper significantly. We are encouraged by the endorsements that:
1) Our manuscript is well written with clear motivation, well-organized flow, and easy-to-understand main results on token dynamics; solid mathematical framework with valid, easily checked analysis (Reviewer waqm, p6VJ).
2) The paper addresses the effect of positional encoding on Transformer token dynamics, an unexplored area, and extends results to more general parameter matrices (Reviewer 1jLS). This line of research is interesting and relevant (Reviewer m1Uq), clearly **fundamental and critical** for improving the interpretability of large language models (Reviewer 1jLS).
3) The theoretical findings are supported by extensive set of experiments (Reviewer m1Uq)

One main concern from reviewers is the simplifying assumption that attention weights are tied across layers. In response, we examine whether our theorems generalize to the setting where model parameters depend on $t$ (see Answer A3 to Reviewer waqm). Extending the proof of Theorem 3.6--one of the main results supporting our experiments-we find it still holds when both $W$ and $V$ vary with $t$, provided $V$ has fixed positive eigenvalues for all $t > 0$.

We also clarify that, while prior works have studied token dynamics in transformers and other deep models, none have addressed the technically challenging case where parameters vary with time. We view this as an important research direction for future work.

Incorporating the comments and suggestions from all reviewers, besides fixing typos and notations, we will incoporate the following main changes in the revised version:

1. Add clarification to proofs and main text, provide more discussions with related works (Response to Reviewer waqm, 1jLS, p6VJ, m1Uq)
2. Improve figure explanations (Figs. 3, 4, 6, 7) and highlight our contributions in the main text. (Response to Reviewer p6VJ, m1Uq)
3. Include more experiments on real-world model statistics to verify the theory. (Response to Reviewer waqm, m1Uq)

In summary, we have fully addressed the initial concerns--both by clarifying results already presented in the manuscript and by conducting extensive new experiments during the rebuttal. We appreciate that **all reviewers found our responses convincing and have addressed their concerns** during the discussion period.

Best regards,

Authors

---

### Note · Authors · 2025-08-13

Dear AC and Reviewers,

Thanks for your thoughtful reviews and valuable comments, which have helped us improve the paper significantly. We are encouraged by the endorsements that:
1) Our manuscript is well written with clear motivation, well-organized flow, and easy-to-understand main results on token dynamics; solid mathematical framework with valid, easily checked analysis (Reviewer waqm, p6VJ).
2) The paper addresses the effect of positional encoding on Transformer token dynamics, an unexplored area, and extends results to more general parameter matrices (Reviewer 1jLS). This line of research is interesting and relevant (Reviewer m1Uq), clearly **fundamental and critical** for improving the interpretability of large language models (Reviewer 1jLS).
3) The theoretical findings are supported by extensive set of experiments (Reviewer m1Uq)

One main concern from reviewers is the simplifying assumption that attention weights are tied across layers. In response, we examine whether our theorems generalize to the setting where model parameters depend on $t$ (see Answer A3 to Reviewer waqm). Extending the proof of Theorem 3.6--one of the main results supporting our experiments-we find it still holds when both $W$ and $V$ vary with $t$, provided $V$ has fixed positive eigenvalues for all $t > 0$.

We also clarify that, while prior works have studied token dynamics in transformers and other deep models, none have addressed the technically challenging case where parameters vary with time. We view this as an important research direction for future work.

Incorporating the comments and suggestions from all reviewers, besides fixing typos and notations, we will incoporate the following main changes in the revised version:

1. Add clarification to proofs and main text, provide more discussions with related works (Response to Reviewer waqm, 1jLS, p6VJ, m1Uq)
2. Improve figure explanations (Figs. 3, 4, 6, 7) and highlight our contributions in the main text. (Response to Reviewer p6VJ, m1Uq)
3. Include more experiments on real-world model statistics to verify the theory. (Response to Reviewer waqm, m1Uq)

In summary, we have fully addressed the initial concerns--both by clarifying results already presented in the manuscript and by conducting extensive new experiments during the rebuttal. We appreciate that **all reviewers found our responses convincing and have addressed their concerns** during the discussion period.

Best regards,

Authors

---

### Decision · Program_Chairs · 2025-09-17

**Decision:**

Accept (poster)

**Comment:**

This paper investigates the impact of different forms of positional encoding upon token representations in Transformers from a theoretical perspective in the continuous-time limit. It shows that the behavior as time goes to infinity is ultimately determined by the positive- or negative-definiteness of two matrices formed from Q, K, and V, assuming that those matrices are symmetric. It further proposes a strategy to ensure divergent behavior rather than convergent behavior.

Reviewers at first found the disconnect between the setting studied and real Transformers to be implausible, especially the weight-tying assumption. In response, the authors pointed to other papers that make similar assumptions and papers that make more restrictive assumptions, and, perhaps more importantly, extended Theorem 3.6 to untied weights. They also pointed to results showing that aspects of the theory seem to hold for real networks, e.g. when LayerNorm and the FFN are present token trajectories still cluster, and the eigenvalue distributions of symmetrized matrices from real models are consistent with the theory.

Ultimately, the reviewers were satisfied, and all reviewers now recommend either borderline acceptance or acceptance. I concur with their assessment.